# Osterix-Cre marks distinct subsets of CD45- and CD45+ stromal populations in extra-skeletal tumors with pro-tumorigenic characteristics

Biancamaria Ricci[1], Eric Tycksen[2], Hamza Celik[3], Jad I Belle[3], Francesca Fontana[3], Roberto Civitelli[4], Roberta Faccio[1,5]*

[1]Department of Orthopedics, Washington University School of Medicine, St. Louis, United States; [2]Genome Technology Access Center, Department of Genetics, Washington University School of Medicine, St. Louis, United States; [3]Department of Medicine, Division of Oncology, Washington University School of Medicine, St. Louis, United States; [4]Department of Medicine, Division of Bone and Mineral Diseases, Washington University School of Medicine, St. Louis, United States; [5]Shriners Children Hospital, St. Louis, United States

**Abstract** Cancer-associated fibroblasts (CAFs) are a heterogeneous population of mesenchymal cells supporting tumor progression, whose origin remains to be fully elucidated. Osterix (Osx) is a marker of osteogenic differentiation, expressed in skeletal progenitor stem cells and bone-forming osteoblasts. We report *Osx* expression in CAFs and by using Osx-cre;TdTomato reporter mice we confirm the presence and pro-tumorigenic function of TdT[OSX]+ cells in extra-skeletal tumors. Surprisingly, only a minority of TdT[OSX]+ cells expresses fibroblast and osteogenic markers. The majority of TdT[OSX]+ cells express the hematopoietic marker CD45, have a genetic and phenotypic profile resembling that of tumor infiltrating myeloid and lymphoid populations, but with higher expression of lymphocytic immune suppressive genes. We find *Osx* transcript and Osx protein expression early during hematopoiesis, in subsets of hematopoietic stem cells and multipotent progenitor populations. Our results indicate that *Osx* marks distinct tumor promoting CD45- and CD45+ populations and challenge the dogma that Osx is expressed exclusively in cells of mesenchymal origin.

*For correspondence:
faccior@wustl.edu

Competing interests: The authors declare that no competing interests exist.

## Introduction

In the past couple of decades the common beliefs of how an incipient tumor grows and progresses to a metastatic stage have drastically changed due to the increasing findings of a tight crosstalk between tumor cells and the surrounding stroma, known as the tumor microenvironment (TME) (*Hanahan and Weinberg, 2011*). Tumor stroma comprises a variety of different cells from the mesenchymal and hematopoietic compartments, with pro- and anti-tumor functions. Cancer-associated fibroblasts (CAFs) are cells of the mesenchymal lineage involved in supporting various stages of tumorigenesis from growth to metastatic dissemination (*Kalluri and Zeisberg, 2006*; *Chen and Song, 2019*). Their abundance and phenotypic markers differ among the various types of cancers. CAFs support tumor growth by stimulating tumor cell proliferation, inhibiting apoptotic signals and anti-tumor immune responses, altering the extra-cellular matrix (ECM) to favor invasiveness, and by offering protection from chemotherapeutic approaches. Their heterogeneity probably accounts for the different tumor promoting effects and reflects the fact that CAFs can originate from multiple cellular sources (*Chen and Song, 2019*). Several studies indicate that CAFs derive from bone marrow

mesenchymal stem and progenitor cells recruited at the tumor site. Additional evidence indicates that they can also derive from myofibroblasts, as well as from the trans-differentiation of local pericytes, endothelial and epithelial cells. In breast cancer, subsets of bone marrow-derived CAFs can exhibit a unique inflammatory profile depending on the location to which they are recruited, thus being functionally distinct from the resident CAFs and having better tumor-promoting functions (*Raz et al., 2018*). Adding more complexity to the origin of CAFs, an additional population expressing several markers commonly found in fibroblasts and CAFs, is the fibrocyte (*Abe et al., 2001*; *McDonald and LaRue, 2012*; *van Deventer et al., 2013*). Fibrocytes originate from the bone marrow, where they have been described to contribute to bone marrow fibrosis under pathological conditions (*Ohishi et al., 2012*). These cells have also been implicated in supporting tumor progression, but they differ from the bone marrow-derived CAF populations in that they also express the hematopoietic marker CD45, along with markers associated with the monocyte/macrophage lineage populations (*Abe et al., 2001*). Fibrocytes support tumor growth by making collagen, albeit at lower levels than the CAFs, releasing growth factors, commonly produced by immune suppressive myeloid cells, and by modulating resistance to anti-angiogenic therapy (*Goto and Nishioka, 2017*). Recent single-cell RNA sequencing (scRNAseq) studies further confirmed the complexity of the CAF populations, suggesting tumor specificity as well as functional differences among the various subsets (*Bartoschek et al., 2018*; *Costa et al., 2018*; *Elyada et al., 2019*). However, these studies did not directly address the origin of the various subsets, which could account for their specific cellular phenotype.

Because of their ability to produce collagen and other extracellular matrix proteins, we sought to determine whether a subset of bone marrow-derived CAFs could share markers of committed osteolineage cells. Osteolineage cells derive from bone marrow mesenchymal stem cells (MSCs) and their differentiation program is driven by sequential activation of two specific transcription factors, *Runx2* and *Sp7(Osx gene)*, and subsequent acquisition of a phenotype characterized by the ability to produce bone matrix, primarily type I collagen, and to mineralize (*Nakashima et al., 2002*). Although Osx is largely thought as a marker of differentiated osteoblasts, emerging data demonstrate that in the embryonic and perinatal bone marrow *Sp7* is expressed in definitive MSCs that give rise to the marrow stroma, including osteoblasts and adipocytes (*Liu et al., 2013*; *Mizoguchi et al., 2014*). Furthermore, during embryogenesis, Osx is present in extra-skeletal tissues, including the olfactory bulb, the intestine and the kidney (*Chen et al., 2014*; *Jia et al., 2015*).

Based on the above observations, we hypothesized that a subset of CAFs, derived from Osx+ cells in the bone marrow, contributes to ECM (i.e. collagen) production at the tumor site, thereby creating a tumor supporting stroma. Using a cell tracking system, we found the presence of cells targeted by the Osx promoter within the TME; these TdT$^{OSX}$+ cells favor tumor growth when co-injected with tumor cells in mice. Surprisingly, only a minority of tumor-resident cells derived from Osx+ cells expresses fibroblast markers, extracellular matrix and matrix remodeling genes. The majority of these newly identified TdT$^{OSX}$+ tumor infiltrating cells are also positive for CD45, a marker of hematopoietic lineage, and share markers expressed by tumor-infiltrating immune cells. Importantly, we confirmed *Sp7* transcripts and Osx protein in a subset of hematopoietic stem cells (HSC), giving rise to TdT$^{OSX}$+;CD45+ tumor infiltrating immune populations. This study further identifies new populations of TME cells targeted by Osx and challenges the use of Osx-cre driven lineage tracing mouse models to exclusively study mesenchymal lineage cell fate.

## Results

### Embryonic and adult-derived osteolineage Osx+ cells are present in extra-skeletal tumors

To determine whether osteolineage cells may be present in the TME, we crossed the established tetracycline-dependent *Tg(Sp7-tTA,tetO-EGFP/cre)1Amc/J* (Osx-cre) to the *B6.Cg-Gt(ROSA)26Sortm9 (CAG-tdTomato)Hze/J* (TdT) to generate the Osx-cre;TdT reporter mouse model (*Rodda and McMahon, 2006*). When constitutively activated, TdT marks the entire osteolineage, including bone surface osteoblasts, osteocytes and bone marrow cells with mesenchymal stem and osteoprogenitor cell features; while delaying Osx-cre expression until postnatally restricts TdT targeting to committed osteoblasts and osteocytes (*Mizoguchi et al., 2014*; *Fontana et al., 2017*). Therefore, Osx-cre;TdT

mice and control animals carrying only the TdT transgene (WT;TdT) were kept on standard chow to allow constitutive embryonic transgene activation, or fed a doxycycline (doxy)-containing diet until weaning to suppress transgene activation until one month of age (*Figure 1A*). We previously reported that doxy-fed mice display less than 1% of spontaneous recombination in the bone residing osteoblasts at weaning, but full transgene activation 1 month thereafter (*Fontana et al., 2017*).

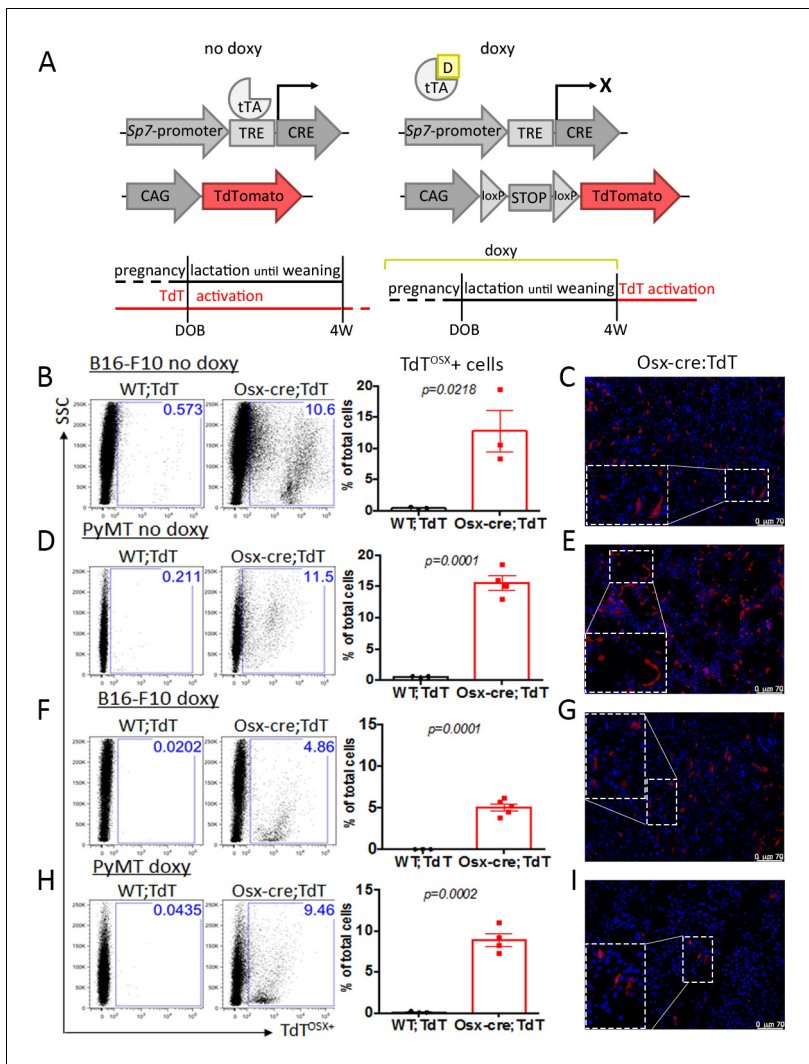

**Figure 1.** Embryonic and adult-derived Osx+ cells are present in primary tumors at extra-skeletal sites. (A) Doxycycline (doxy)-repressible Sp7-cre/loxP mouse model used to activate Ai9/TdTomato expression for lineage tracing experiments. In no doxy-fed mice, TdT is expressed in embryonic-derived osteolineage cells (left), while in mice fed a doxy diet until weaning, TdT is expressed in adult-derived osteolineage cells. (B–I) Flow cytometry analysis and fluorescence images of primary tumors showing presence of TdT$^{OSX}$+ cells in no-doxy fed Osx-cre; TdT mice or WT;TdT controls inoculated with B16-F10 melanoma subcutaneously (B–C), or with PyMT breast cancer cells in the mammary fat pad (MFP) (D–E), and in doxy-fed mice injected with B16-F10 subcutaneously (F–G), or with PyMT in the MFP (H–I). Slides for fluorescence images were counterstained with DAPI (blue), magnification 200X. Inserts are further magnified 4.5 folds. Data representative of a single experiment, experiments were repeated between 2 and 5 times. p values represent Student t-test statistical analysis.
The online version of this article includes the following source data and figure supplement(s) for figure 1:

**Source data 1.** Relates to FACS analysis in panels B, D, F, H.
**Figure supplement 1.** No difference in tumor growth and body weight between sex and age matched cre positive (Osx-cre;TdT) and negative (WT;TdT) mice.

Osx-cre;TdT reporter mice and WT;TdT fed a normal diet (no doxy) were inoculated with either $10^5$ B16-F10 melanoma cells subcutaneously or $10^5$ PyMT breast cancer cells in the mammary fat pad (MFP). To account for the possible growth delay sometimes observed in no-doxy fed Osx-cre mice (*Davey et al., 2012*), tumor cells were injected at 12 weeks of age, when the growth delay is fully recovered. Importantly, no differences in tumor growth were observed in age and sex matched Osx-cre;TdT and WT;TdT mice, excluding any potential confounding effect of the Osx-cre (*Figure 1—figure supplement 1*). To assess the presence of osteolineage Osx+ derived cells within the tumor stroma of no doxy-fed mice, we performed flow cytometry analysis on B16-F10 and PyMT tumors isolated 2 weeks post-inoculation, and found that about 10–18% of the total cells were TdT$^{OSX}$+ (*Figure 1B and D*). Histological analysis of frozen sections from the same tumors confirmed the presence of TdT$^{OSX}$+ cells. These cells had either round or elongated shape, indicating morphologic heterogeneity within the TdT$^{OSX}$+ population (*Figure 1C and E*). These TdT$^{OSX}$+ cells within the tumor stroma of no-doxy mice may derive from resident Osx-expressing embryonic progenitors or from mobilization of bone marrow Osx+ cells.

Remarkably, TdT$^{OSX}$+ cells were also present in the B16-F10 and PyMT TME of doxy-fed animals 4 weeks post weaning, although their numbers were lower than in tumors isolated from no doxy-treated mice (*Figure 1F and H*). Fluorescence micrographs also confirmed the presence of TdT$^{OSX}$+ cells in the stroma in both tumor models (*Figure 1G and I*), with the same pleiotropic morphology as seen in no doxy-treated mice. Thus, embryonic and adult-derived osteolineage Osx+ cells are present in the stroma of extra-skeletal tumors. Based on these findings, the doxy-fed Osx-cre;TdT mice were used for all the subsequent studies.

## TdT$^{OSX}$+ cells from primary tumors express mesenchymal markers

To determine the phenotype of TdT$^{OSX}$+ cells more in depth, we isolated TdT$^{OSX}$+ cells by FACS sorting 14 days after inoculation of B16-F10 tumor cells into doxy-fed Osx-cre;TdT reporter mice (*Figure 2A*). Due to the limited number of TdT$^{OSX}$+, tumors from 2 to 4 mice were pooled for FACS sorting and reported as one experimental replicate (single data point). Messenger RNA from an immortalized murine CAF cell line (i-CAFs) and cortical long bone devoid of bone marrow cells were used as controls. Real-Time PCR confirmed *Sp7* expression in TdT$^{OSX}$+ cells but not in the TdT$^{OSX}$– fraction, which includes the remaining stroma and the tumor cells (*Figure 2B*). Interestingly, *Sp7* expression was also detected in i-CAFs (*Figure 2B*), and as expected in bone extracts in higher abundance. Expression of osteoblast-specific markers such as *Runx2*, *Bglap (Ocn)* (Osteocalcin), *Ibsp (Bsp, bone sialoprotein)* and *Alpl (Tnap, tissue non-specific alkaline phosphatase)* were negligible in TdT$^{OSX}$+ cells and undetectable or barely detectable in the i-CAFs (*Figure 2C–F*). However, *S100a4 (Fsp1)* (Fibroblast-specific protein 1) and *Acta2 (α-SMA)* (α-smooth muscle actin), two CAFs specific markers, were detected in TdT$^{OSX}$+ cells (*Figure 2G and H*). TdT$^{OSX}$+ also expressed *Col1a1* (Collagen Type Ia1) and *Col1a2* (Collagen Type Ia2) similar to i-CAFs and bone extracts (*Figure 2I and J*). Expression of matrix metalloproteinases, *Mmp2* and *Mmp9*, involved in matrix remodeling and highly expressed in CAFs and osteoblasts respectively, was also detected in the TdT$^{OSX}$+ cells (*Figure 2K and L*).

To validate expression of *Sp7* in CAFs from primary tumors, we orthotopically injected B16-F10 or PyMT cells in 8 week old WT mice and the CAFs were isolated as the adherent fraction of a single cell suspension of the tumor mass. We detected *Sp7* expression only in CAFs but not in the tumor cell lines nor in the non-adherent fraction of the single cell suspension from the tumor mass (*Figure 2M and N*). Importantly, corroborating our findings in mice, using Oncomine (*Rhodes et al., 2004*) we found that microarray analysis of human breast stromal cells collected from patients with invasive breast carcinoma from the Finak gene set (*Finak et al., 2008*) showed significantly higher *Sp7* expression in the tumor associated stroma compared to the adjacent normal breast tissue (*Figure 2O*). Thus, a subset of cells in the TME of mice and humans express *Sp7*, an osteolineage cell marker, but also markers associated with the CAFs. TdT$^{OSX}$+ cells may represent a subpopulation of CAFs with specific features.

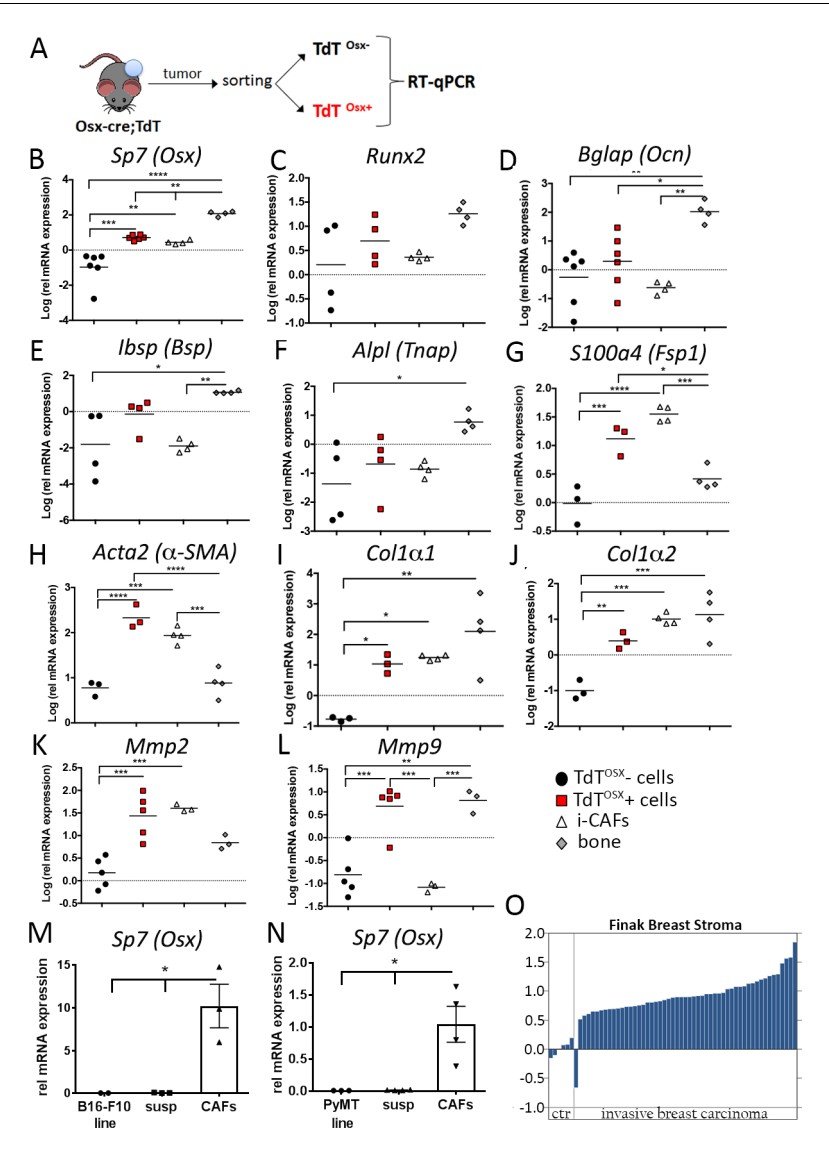

**Figure 2.** TdT[OSX]+ cells from primary tumors express mesenchymal markers. (A) Experimental design for isolating TdT[OSX]+ cells from primary tumors isolated from doxy-fed Osx-cre;TdT mice inoculated with B16-F10. TdT[OSX]- cells comprise all TdT negative cells including the tumor cells. Immortalized CAFs (iCAFs) and crushed long bones were used as reference controls. (B–L) Quantitative Real-Time PCR for (B) *Sp7(Osterix)*, (C) *Runx2*, (D) *Bglap (Osteocalcin)*, (E) *Ibsp (Bone sialoprotein)*, (F) *Alpl (Tnap, Tissue non-specific alkaline phosphatase)*, (G) *S100a4 (Fsp1)*, (H) *Acta2 (α-SMA)* (I) *Collagen1a1* (J) *Collagen1a2*, (K) *Mmp2 (Metalloproteinase2)* and (L) *Mmp9 (Metalloproteinase9)* genes in TdT[OSX]+, TdT[OSX]- cells, iCAFs and bone. Each data point includes cells isolated from 3 to 4 mice and from two independent experiments. (M–N) Real-Time PCR of *Sp7(Osterix)* in B16-F10 or PyMT cell lines, CAFs isolated from primary B16-F10 or PyMT tumors or from the remaining cells comprising the tumor minus the CAFs (susp) (n = 3–4 mice). (O) *Sp7(Osx)* expression based on the Finak Breast Cancer Stroma gene set in the Oncomine cancer microarray database. Significance was determined by one-way ANOVA with Tukey post-hoc test, *p<0.05, **p<0.01, ***p<0.001, ****p<0.0001.

The online version of this article includes the following source data for figure 2:

**Source data 1.** Relates to Real-Time PCR data.

## Tumor resident but not bone marrow resident TdT^OSX+ cells increase tumor growth

To determine whether tumor resident TdT$^{OSX}$+ cells can affect tumor growth as do CAFs, TdT$^{OSX}$+ cells were sorted from primary B16-F10 tumors inoculated in 8 week old doxy-fed Osx-cre;TdT mice and re-injected along with B16-F10 tumor cells at 5:1 ratio into age-matched WT recipient animals (*Figure 3A*). Remarkably, co-injecting TdT$^{OSX}$+ with tumor cells resulted in larger tumors (*Figure 3B*) compared to injection of B16-F10 cells alone.

Because TdT$^{OSX}$+ cells normally reside in the bone microenvironment (*Fontana et al., 2017*; *Strecker et al., 2013*), we next examined bone sections from 8 week old doxy-fed Osx-cre;TdT reporter mice, naïve or bearing soft tissue B16-F10 or PyMT tumors. We confirmed presence of TdT$^{OSX}$+ cells on the surface of cortical and trabecular bone, TdT$^{OSX}$+ osteocytes, and also abundant staining in the bone marrow. TdT was not detected WT;TdT mice (*Figure 3C* and *Figure 3—figure supplement 1A*). We further analyzed the percentage of TdT$^{OSX}$+ cells in the bone marrow by FACS and found that TdT$^{OSX}$+ cells accounted for about 10% of total marrow cells in naïve mice and their number increased in mice injected with B16-F10 or PyMT tumor cells (*Figure 3D* and *Figure 3—figure supplement 2B*). Similar findings were observed in the Tamoxifen (TAM) inducible *Sp7-creER^{T2}; Rosa26^{<fs-TdTomato>}* (Osx-creER^{T2};TdT) mouse model. TAM was administered to 7 week old mice via IP injections starting 3 days prior to B16-F10 tumor inoculation, and 1, 6, and 9 days post tumor cell injection (*Figure 4—figure supplement 1A*). No tumor bearing mice were used as control. Similar to the Tet-OFF Osx-cre model, we found that bone marrow TdT$^{OSX}$+ cells significantly increased in presence of a tumor (*Figure 3E*).

To test their capacity to promote tumor growth, bone marrow-derived TdT$^{OSX}$+ cells were isolated from tumor bearing mice and re-injected with B16-F10 tumor cells into WT mice (5:1 ratio). In contrast to what observed with tumor-derived TdT$^{OSX}$+ cells, bone marrow-derived TdT$^{OSX}$+ cells did not enhance tumor growth compared to mice injected with tumor cells alone; if anything, there was a trend towards a smaller tumor size (*Figure 3F*). This result further prompted us to isolate bone marrow-derived TdT$^{OSX}$+ cells and analyze the expression levels of the same mesenchymal genes analyzed in the tumor-derived TdT$^{OSX}$+ cells (*Figure 2*). Interestingly, TdT$^{OSX}$+ cells from the TME expressed higher levels of *Sp7*, CAF markers *S100a4* and *Acta2*, and matrix proteins *Col1a1* and *Ibsp* compared to TdT$^{OSX}$+ from the bone marrow (*Figure 3—figure supplement 2*). Bone marrow TdT$^{OSX}$+ cells instead expressed significantly higher levels of the early osteoblast transcription factor *Runx2* and of the protease *Mmp9*, while no differences in other osteoblast markers such as *Bglap* and *Alpl* were detected between the tumor and bone marrow sorted TdT$^{OSX}$+ cells (*Figure 3—figure supplement 2*).

These results suggest that Osx+ cells infiltrating a tumor are functionally and phenotypically distinct from bone marrow resident Osx+ cells.

## The majority of TdT^OSX+ cells express CD45

To gain insights on the possible origin of tumor-derived TdT$^{OSX}$+ cells, we performed FACS analysis of circulating cells in naïve mice and two weeks post B16-F10 or PyMT tumor inoculation. Interestingly, we found 5–8% of TdT$^{OSX}$+ cells in the blood of tumor-free doxy-fed Osx-cre;TdT mice, and this percentage was 3- to 4-fold higher in tumor-bearing animals (*Figure 4A and B*). The unexpected high number of circulating TdT$^{OSX}$+ cells prompted us to determine whether they may express the immune marker CD45, which is also present in fibrocytes. Surprisingly, 95% of TdT$^{OSX}$+ cells in circulation of tumor bearing mice were also positive for CD45, and they represented about 13–18% of total blood cells (*Figure 4C and D*). Similarly, about 92–95% of total TdT$^{OSX}$+ cells in the bone marrow were CD45+ (*Figure 4E and F*), representing 12–20% of total marrow cells. Such finding explained the low expression levels of mesenchymal markers in the bone marrow-derived TdT$^{OSX}$+ cells (*Figure 3—figure supplement 2*). Intriguingly, also the majority of TdT$^{OSX}$+ cells in the TME expressed CD45 (*Figure 4G and H*), data further confirmed by RT-PCR in TdT$^{OSX}$+ fraction sorted from B16-F10 tumors (not shown).

To exclude a possible off-target effect of the doxy-dependent Tet-OFF system, we turned to the TAM inducible Osx-creER^{T2};TdT model. FACS analysis confirmed presence of both CD45- and CD45+ TdT$^{OSX}$+ populations in the bone marrow of B16-F10 bearing Osx-creER^{T2};TdT mice, but not in the cre negative mice (*Figure 4—figure supplement 1B*).

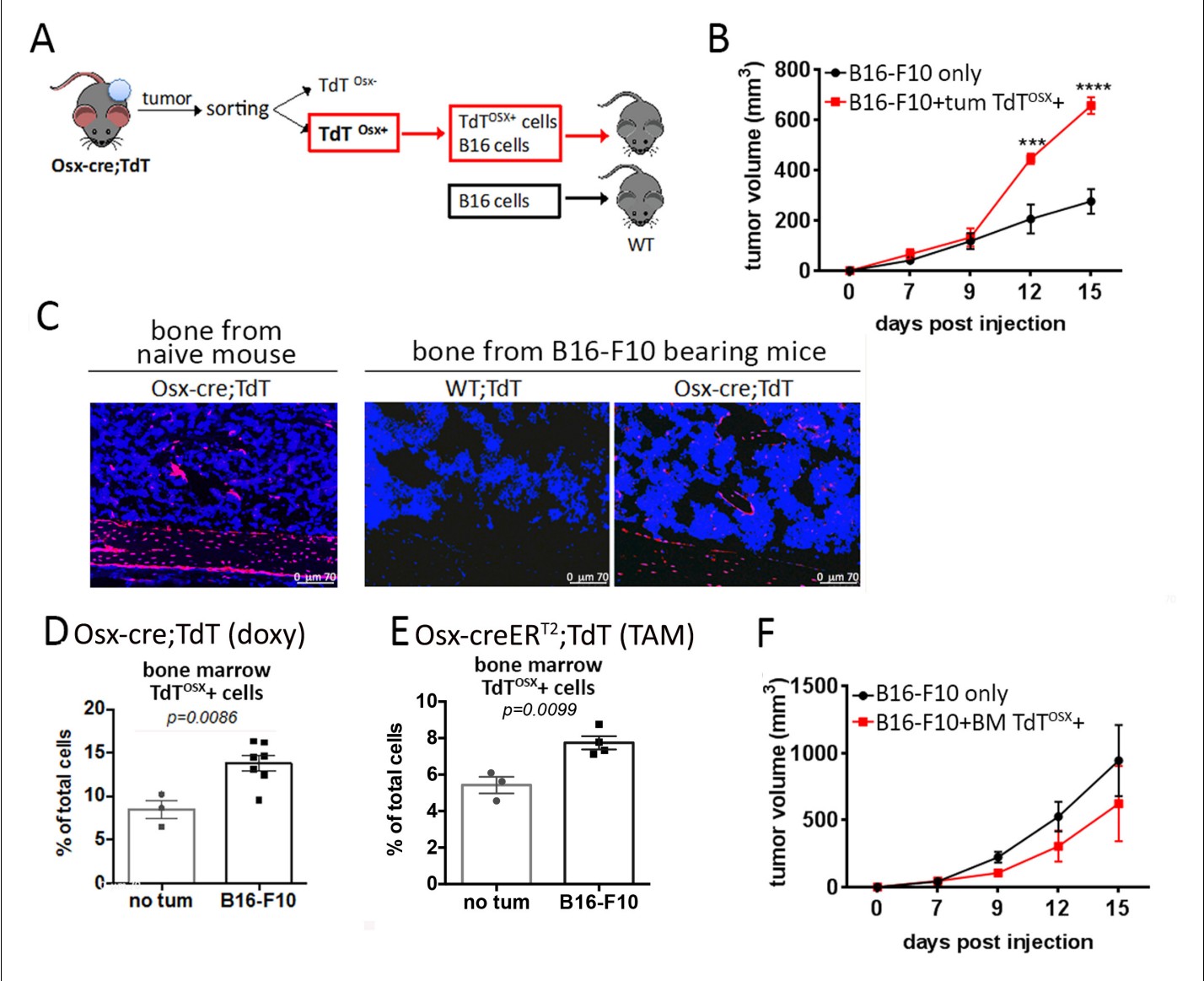

**Figure 3.** Tumor resident but not bone marrow resident TdT^OSX+ cells increase tumor growth. (**A**) Model showing isolation of TdT^OSX+ cells from B16-F10 primary tumors injected into doxy-fed Osx-cre;TdT mice and re-inoculation of TdT^OSX+ cells together with B16-F10 tumor cells into WT recipient mice at the ratio 5:1. Mice injected with B16-F10 alone were used as controls. (**B**) Tumor growth of experimental model described in (**A**) determined by caliper measurement. n = 3/group, experiment repeated twice. Significance was determined by one-way ANOVA with Tukey post-hoc test. (**C**) Fluorescence images of bone sections showing presence of TdT^OSX+ within the bone and bone marrow of naïve Osx-cre;TdT, Osx-cre;TdT mice inoculated subcutaneously with B16-F10 tumors or WT;TdT mice used as negative control. Sections were counterstained with DAPI (blue), magnification 200X. (**D–E**) Quantification of TdT^OSX+ cells in the bone marrow of tumor-free and B16-F10 tumor bearing (**D**) Osx-cre;TdT mice (doxy-fed) and (**E**) Osx-creER^T2;TdT (TAM-treated) determined by FACS. Experiment repeated twice. Significance was determined by student t-test statistical analysis. (**F**) Tumor growth of WT mice inoculated with bone marrow-derived TdT^OSX+ cells from B16-F10 bearing doxy-fed Osx-cre;TdT mice together with B16-F10 tumor cells (ratio 5:1). Mice injected with B16-F10 alone were used as controls. n = 3/group. Significance was determined by two-way ANOVA followed by Tukey post-hoc test. ***p<0.001, ****p<0.0001.

The online version of this article includes the following source data and figure supplement(s) for figure 3:

**Figure supplement 1.** TdT^OSX+ cells increase in the bone marrow of PyMT tumor bearing mice.

**Figure supplement 1—source data 1.** Relates to FACS analysis.

**Figure supplement 2.** Differential expression of mesenchymal markers in TdT^OSX+ cells isolated from tumor site or bone marrow.

**Figure supplement 2—source data 1.** Relates to Real-Time PCR.

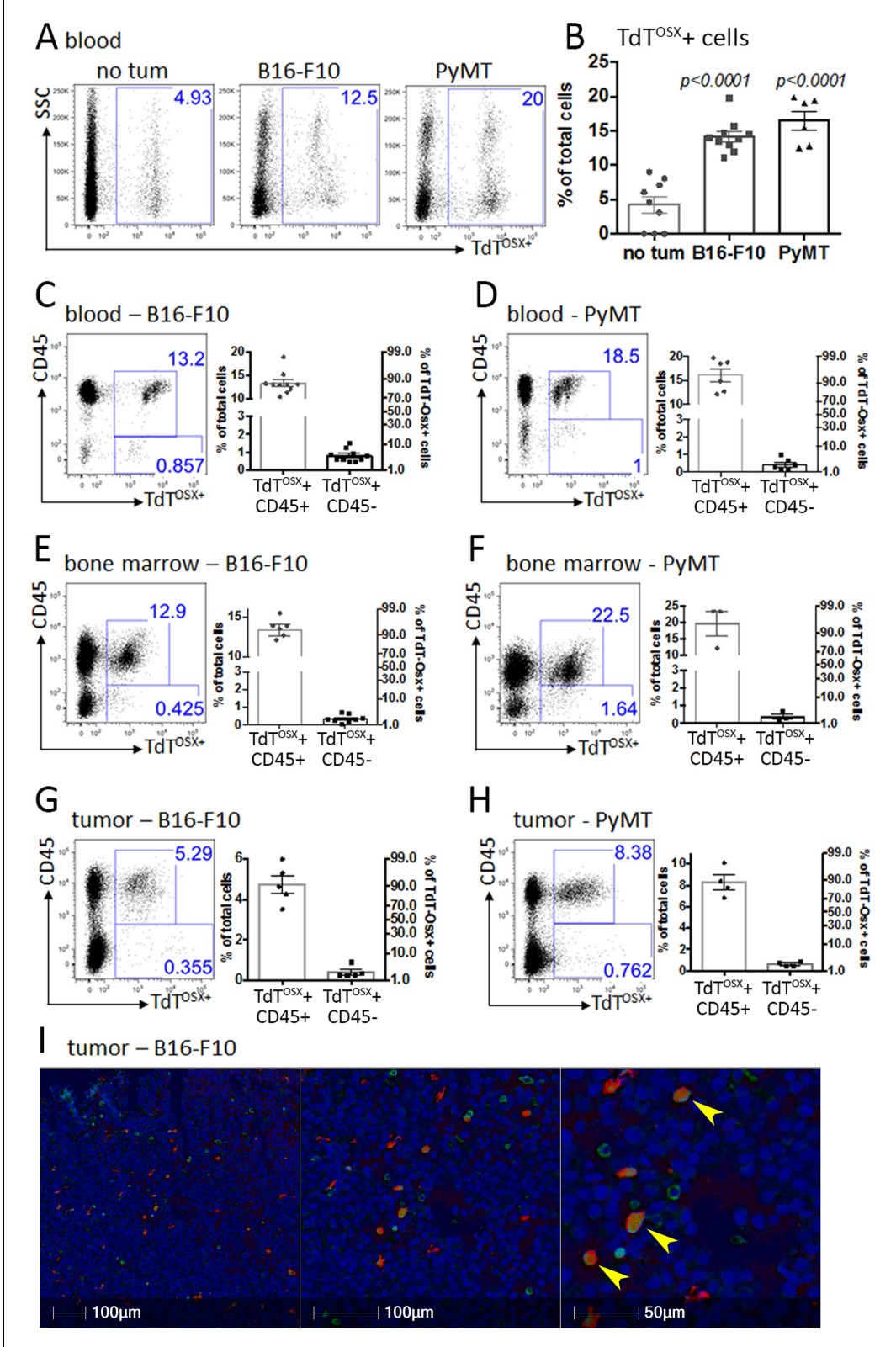

**Figure 4.** TdT^OSX+ cells express the immune surface marker CD45. (**A–B**) Representative FACS dot plots and quantification of TdT^OSX+ cells in the peripheral blood of tumor-free and B16-F10 or PyMT tumor bearing mice. Significance was determined by student t-test statistical analysis for each tumor model relative to no tumor controls. (**C–D**) Representative FACS dot plots and quantification of TdT^OSX+;CD45+ and TdT^OSX+;CD45-populations in the blood of doxy-fed Osx-cre;TdT mice injected with B16-F10 or PyMT cells. (**E–H**) Representative FACS dot plots and quantification of

*Figure 4 continued on next page*

*Figure 4 continued*

TdT^OSX^+;CD45+ and TdT^OSX^+;CD45- populations in the bone marrow (**E–F**) or tumor site (**G–H**) of doxy-fed Osx-cre;TdT mice injected with B16-F10 or PyMT cells, respectively. Experiments were repeated at least twice. (**I**) Immunohistochemistry staining of paraffin-embedded B16-F10 tumors inoculated into Osx-cre;TdT reporter mice, pseudo colored with red representing TdT^OSX^+ cells (RFP stained), green representing CD45+ cells (DAB stained) and blue representing nuclei (hematoxylin), magnification 200X.

The online version of this article includes the following source data and figure supplement(s) for figure 4:

**Source data 1.** Relates to FACS analysis in panels B, C, D, E, F, G, H.
**Figure supplement 1.** Presence of TdT^OSX^+ cells in bone marrow (BM) of TAM-pulsed Osx-creER^T2^;TdT reporter mice.
**Figure supplement 1—source data 1.** Relates to FACS analysis.
**Figure supplement 2.** Co-expression of Osx-driven TdTomato and the immune marker CD45.

Next, to confirm that expression of CD45 in the TdT^OSX^+ cells was not due a tight interaction between a CD45+ immune cell and TdT^OSX^+ mesenchymal cell, we performed CD45 immunostaining on paraffin-embedded B16-F10 tumor sections and on bone marrow single cell suspension from Osx-cre;TdT reporter mice. We found co-localization of Osx-driven TdTomato and CD45 in the same cell, confirming that CD45 is indeed expressed in subsets of TdT^OSX^+ cells (*Figure 4I* and *Figure 4— figure supplement 2*). Thus, Osx not only marks mesenchymal cells, but also cells expressing the hematopoietic marker CD45, suggesting that they could represent bone marrow fibrocytes or an immune cell subset.

## Presence of functionally distinct populations of TdT^OSX^+ cells in the tumor microenvironment

To further characterize the phenotype of the CD45- and CD45+ TdT^OSX^+ populations, we next performed RNAseq analysis. We subcutaneously injected the GFP-labeled PyMT-BO1-tumor line in Osx-cre;TdT reporter mice. Two weeks post tumor inoculation, the stromal cells were separated from the GFP+ tumor cells and sorted based on the expression, or lack of thereof, of TdT and CD45 markers. We sorted 4 groups of cells: double negative (TdT^OSX^-;CD45-), representing the non-immune tumor stroma, CD45 single positive (TdT^OSX^-;CD45+), representing the tumor infiltrating immune populations, TdT^OSX^ single positive (TdT^OSX^+;CD45-), and double positive (TdT^OSX^+;CD45+).

Principal components analysis of RNA-seq expression patterns across all four groups revealed clustering that was uniquely dependent on CD45 expression and suggested that there were only two very distinct cell populations regardless of TdT^OSX^ (*Figure 5A*). Inspection of each cell population in a one versus all other approach for only statistically significant up-regulated genes (FDR >= 0.05, log2 fold-change >= 2) to identify robustly expressed biomarkers, showed that all four groups shared many genes in common based on CD45 expression alone. The CD45 negative populations shared 1248 genes in common with 536 in the TdT^OSX^ single positive and 477 in the double negative cells uniquely up-regulated (*Figure 5B*). Likewise, the CD45 single and double positive cells shared 495 significantly up-regulated genes with an additional 77 and 316 uniquely expressed, respectively.

Subsequent Gene Ontology (GO) analysis of the $\log_2$ fold-changes for each cell population revealed that the CD45 negative subsets showed upregulation of GO pathways related to cellular responses to growth factors, extracellular structure and matrix organization and skeletal system development, confirming their mesenchymal nature (*Figure 5C*). Among the top upregulated genes in the TdT^OSX^ single positive cells versus the double negative (*Table 1*) we found several markers expressed by osteolineage cells, such as fibromodulin (*Fmod*), a binding protein regulating bone mineralization (*Gori et al., 2001*) and also involved in TGF$\beta$ signaling during cancer pathogenesis (*Pourhanifeh et al., 2019*), Collagen 24a1 (*Col24a1*), known to modulate collagen chain trimerization (*Koch et al., 2003*; *Matsuo et al., 2006*), *Frem1*, a protein involved in the formation and organization of basement membranes (*Vissers et al., 2011*), and *Msx1*, a transcription factor important in craniofacial bone development (*Orestes-Cardoso et al., 2002*). TdT^OSX^ single positive cells also expressed high levels of fibulin 7 (*Fbln7*), a cell adhesion molecule overexpressed in glioblastoma by pericytes and involved in neovascularization (*de Vega et al., 2019*; *Ikeuchi et al., 2018*). Such result confirmed the osteolineage nature of the TdT^OSX^+ cells and indicated their tumor supporting role.

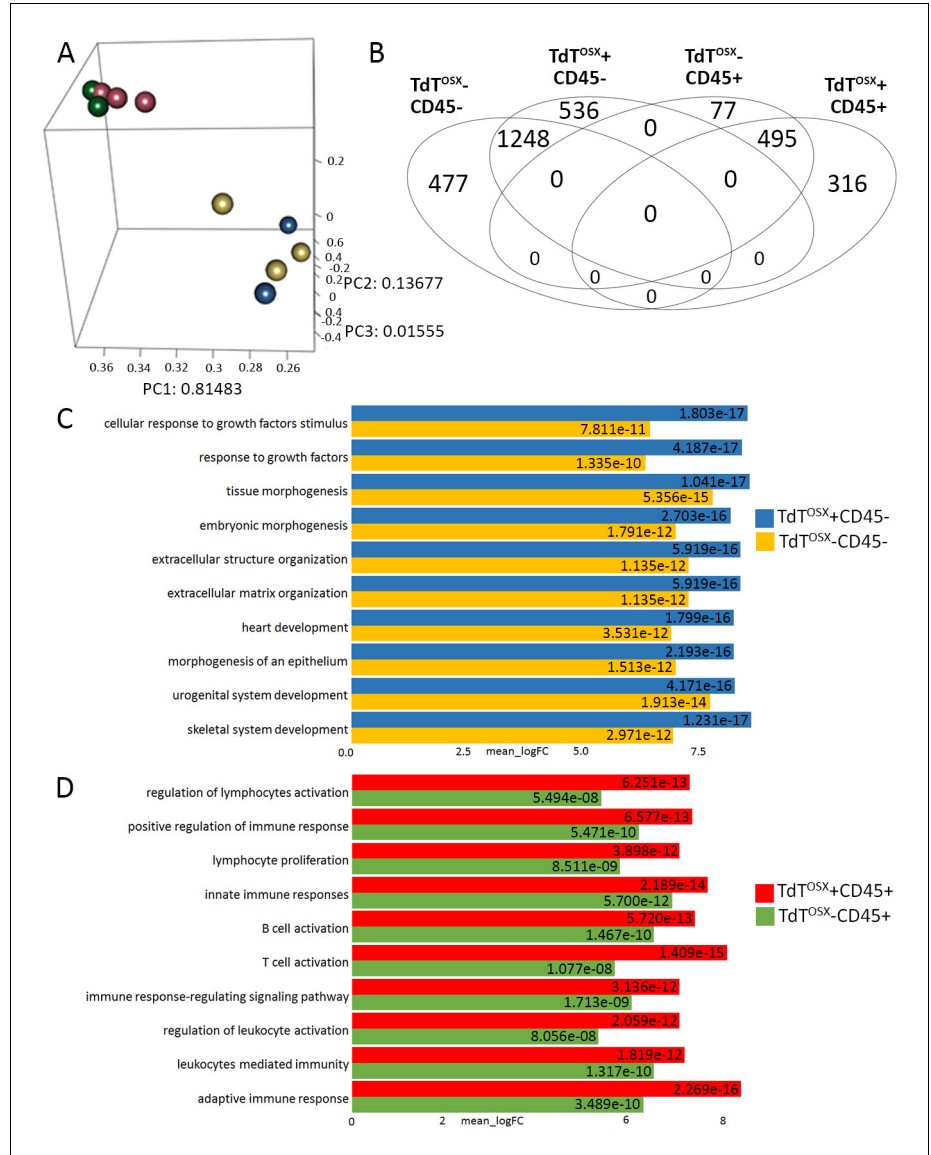

**Figure 5.** TdT$^{OSX}$+;CD45- and TdT$^{OSX}$+;CD45+ are two functionally distinct populations in the tumor microenvironment. (**A**) Multidimensional Principal Component Analysis of RNAseq data obtained from four tumor stroma subsets sorted according to TdT$^{OSX}$ and CD45 expression. (**B**) Venn diagram depicting uniquely and commonly expressed genes among the four groups. (**C–D**) First ten GO pathways obtained from the GO analysis of the log two fold-changes for (**C**) CD45 negative subsets and (**D**) CD45 positive subsets.

The online version of this article includes the following source data and figure supplement(s) for figure 5:

**Figure supplement 1.** Validation of RNAseq data.

**Figure supplement 1—source data 1.** Relates to Real-Time PCR data.

Conversely, both CD45+ subsets were significantly up-regulated for GO pathways related to regulation of lymphocyte activation and proliferation, and innate immune cell responses (*Figure 5D*). Interestingly, among the top genes upregulated in the TdT$^{OSX}$+ CD45+ subsets versus the CD45 single positive populations (*Table 2*), we found many genes expressed by tumor promoting immune cells, such as regulatory T cells (i.e. *Foxp3*) and gamma-delta T cells (i.e. *Tcrg-C1*). The most upregulated gene in the TdT$^{OSX}$+;CD45+ subset (>80 fold versus CD45 single positive) was *Cd163l1*, a marker of gamma-delta IL-17 producing cells (*Tan et al., 2019*) and M2 macrophages (*González-Domínguez et al., 2015*), two tumor-promoting immune populations. We confirmed *Cd163l1* higher expression by RT-PCR together with other two genes, *Wdr78* and *Spock2*, also upregulated in the

**Table 1.** Top 50 uniquely up-regulated genes in tumor-derived TdT^OSX+;CD45- cells.

List of the 50 most highly expressed genes in the TdT^OSX+;CD45- cells compared to the TdT^OSX-;CD45- cells isolated from the TME.

| external_gene_name | description | TdTOSX +CD45- logFC | TdTOSX +CD45- linearFC |
|---|---|---|---|
| Reg1 | regenerating islet-derived 1 | 7.675801 | 204.477899 |
| Fbln7 | fibulin 7 | 7.184048 | 145.416614 |
| Adcyap1r1 | adenylate cyclase activating polypeptide 1 receptor 1 | 6.864422 | 116.51905 |
| Col24a1 | collagen, type XXIV, alpha 1 | 6.788179 | 110.521192 |
| Moxd1 | monooxygenase, DBH-like 1 | 6.74623 | 107.35382 |
| Fmod | fibromodulin | 6.739645 | 106.864989 |
| 4930562D21Rik | RIKEN cDNA 4930562D21 gene | 6.149924 | 71.008705 |
| Angpt4 | angiopoietin 4 | 6.131627 | 70.113799 |
| Tmem132c | transmembrane protein 132C | 5.933002 | 61.095844 |
| Frem1 | Fras1 related extracellular matrix protein 1 | 5.906166 | 59.969876 |
| Cxcl5 | chemokine (C-X-C motif) ligand 5 | 5.88881 | 59.252728 |
| Ccl19 | chemokine (C-C motif) ligand 19 | 5.748355 | 53.756054 |
| Rab15 | RAB15, member RAS oncogene family | 5.696057 | 51.842269 |
| Prlr | prolactin receptor | 5.673651 | 51.04334 |
| Msx1 | msh homeobox 1 | 5.657661 | 50.480738 |
| Pabpc4l | poly(A) binding protein, cytoplasmic 4-like | 5.657301 | 50.46814 |
| Reg3g | regenerating islet-derived 3 gamma | 5.651841 | 50.2775 |
| Rflna | refilin A | 5.651765 | 50.274843 |
| Adamts3 | a disintegrin-like and metallopeptidase (reprolysin type) with thrombospondin type 1 motif, 3 | 5.52646 | 46.092506 |
| Lrp3 | low density lipoprotein receptor-related protein 3 | 5.465865 | 44.196651 |
| Clstn2 | calsyntenin 2 | 5.41628 | 42.703438 |
| Adam5 | a disintegrin and metallopeptidase domain 5 | 5.36495 | 41.210775 |
| H2-M11 | histocompatibility 2, M region locus 11 | 5.335531 | 40.378937 |
| Ptx3 | pentraxin related gene | 5.320565 | 39.962237 |
| Rtl3 | retrotransposon Gag like 3 | 5.252916 | 38.131634 |
| Gm26682 | predicted gene, 26682 | 5.252803 | 38.128648 |
| Morc1 | microrchidia 1 | 5.251594 | 38.096689 |
| Hoxc6 | homeobox C6 | 5.176329 | 36.160165 |
| Mcpt8 | mast cell protease 8 | 5.146576 | 35.422062 |
| Gabrb3 | gamma-aminobutyric acid (GABA) A receptor, subunit beta 3 | 5.110873 | 34.556218 |
| Sbspon | somatomedin B and thrombospondin, type 1 domain containing | 5.085426 | 33.952042 |
| Akap6 | A kinase (PRKA) anchor protein 6 | 5.073102 | 33.663235 |
| Hmcn1 | hemicentin 1 | 5.060346 | 33.366905 |
| Aqp2 | aquaporin 2 | 5.031264 | 32.701036 |
| Gria3 | glutamate receptor, ionotropic, AMPA3 (alpha 3) | 5.011924 | 32.265589 |
| AI464131 | expressed sequence AI464131 | 5.001414 | 32.03138 |
| Muc13 | mucin 13, epithelial transmembrane | 4.997132 | 31.936452 |
| Syt4 | synaptotagmin IV | 4.994666 | 31.881898 |
| Spon1 | spondin 1, (f-spondin) extracellular matrix protein | 4.916033 | 30.190712 |
| Caln1 | calneuron 1 | 4.85996 | 29.039801 |
| Hoxc4 | homeobox C4 | 4.844155 | 28.723415 |
| Tbx2 | T-box 2 | 4.842042 | 28.681376 |

*Table 1 continued on next page*

Table 1 continued

| external_gene_name | description | TdTOSX +CD45- logFC | TdTOSX +CD45- linearFC |
|---|---|---|---|
| Jph2 | junctophilin 2 | 4.816833 | 28.18456 |
| Frzb | frizzled-related protein | 4.79214 | 27.706253 |
| Galnt5 | polypeptide N-acetylgalactosaminyltransferase 5 | 4.77782 | 27.432606 |
| Gm29100 | predicted gene 29100 | 4.77255 | 27.332593 |
| Tspyl5 | testis-specific protein, Y-encoded-like 5 | 4.771509 | 27.312867 |
| Spag17 | sperm associated antigen 17 | 4.770653 | 27.296671 |
| Ldoc1 | regulator of NFKB signaling | 4.7479 | 26.86954 |
| 4833422C13Rik | RIKEN cDNA 4833422C13 gene | 4.730678 | 26.550693 |

double positive cells (*Figure 5—figure supplement 1*). The second most expressed gene was Lymphotoxin beta (*Ltb*), a cytokine produced by lymphocytes and NK cells and associated with carcinogenesis (*Wolf et al., 2010*). The third one, *Klrg1*, negatively regulates cytotoxic lymphocytes and is associated with lymphocyte senescence and dysfunction (*Henson and Akbar, 2009*). Thus, these analyses revealed that two main transcriptional programs predominate based on CD45 expression, and suggest that TdT$^{OSX}$+;CD45- comprise a subset of CAFs with some characteristics of skeletal cells, while TdT$^{OSX}$+;CD45+ cells represent a heterogeneous subset of tumor-promoting immune cells.

## Double positive TdT$^{OSX}$+;CD45+ cells are a heterogeneous immune population enriched in lymphoid cells

Next we performed flow cytometry analysis to better characterize the TdT$^{OSX}$+;CD45+ population and validate some of the RNAseq findings using the B16-F10 melanoma model. We confirmed that the double positive TdT$^{OSX}$+;CD45+ represented about 20% of the total CD45+ cells in both the primary tumor and the bone marrow of doxy-fed Osx-cre;TdT mice (*Figure 6A and B*). Next, we analyzed the percentage of TdT$^{OSX}$+ expressing the common myeloid and lymphoid markers (gate strategy in *Figure 6—figure supplement 1*), such as CD11b (monocytes), F4/80 (macrophages), Gr1 (granulocytes/neutrophils), CD3 (T cells, further divided into CD4+ and CD8+), and NK1.1 (Natural Killer cells). To standardize the comparison between the TdT$^{OSX}$+;CD45+ and TdT$^{OSX}$-;CD45+ (single positive) populations within each sample, data were represented as percentage of either total TdT$^{OSX}$+;CD45+ or total CD45 single positive cells (*Figure 6C and D*). TdT$^{OSX}$+;CD45+ subset expressed both myeloid and lymphoid markers, with a similar pattern as the CD45 single positive cells. In both subsets, the majority of the tumor immune infiltrate included myeloid populations, which represented about 60% of the TdT$^{OSX}$+;CD45+ and over 70% of the total CD45 single positive cells. Lymphoid cells appeared to be more abundant among the TdT$^{OSX}$+;CD45+ (32.3 ± 6.5%), compared to the TdT$^{OSX}$-;CD45+ cells (23.3 ± 4.3%). Based on this observation, we calculated the ratio of lymphoid over myeloid populations and observed that the tumor-infiltrating TdT$^{OSX}$+;CD45 + double positive population was enriched in lymphoid cells (*Figure 6E*). Flow cytometric analysis of the bone marrow from tumor bearing Osx-cre;TdT mice revealed similar pattern of distribution between the TdT$^{OSX}$+;CD45+ and the CD45 single positive immune populations (*Figure 6F and G*), although the frequencies of each immune subset in the bone marrow differed compared to the cells at tumor site. These data suggest that Osx marks multiple immune cell types but that TdT$^{OSX}$+-derived immune cells infiltrating a tumor are skewed towards lymphoid populations.

Since adoptive transfer of tumor-derived TdT$^{OSX}$+ cells increases tumor growth (*Figure 3B*), we asked whether TdT$^{OSX}$+;CD45+ immune populations contribute to tumor progression. We isolated by FACS sorting TdT$^{OSX}$+;CD45+ and the TdT$^{OSX}$-;CD45+ from B16-F10 subcutaneous tumors from doxy-fed Osx-Cre;TdT mice. Sorted cells were then re-injected along with B16-F10 tumor cells (ratio 5:1) into new WT recipient mice. The co-injection of CD45 single positive cells decreased tumor growth relative to B16-F10 injected alone at day 15, while TdT$^{OSX}$+;CD45+ did not show anti-tumor effects (*Figure 6H*).

**Table 2.** Top 50 uniquely up-regulated genes in tumor-derived TdT$^{OSX}$+;CD45+ cells.

List of the 50 most highly expressed genes in the TdT$^{OSX}$+;CD45+ cells compared to the TdT$^{OSX}$-;CD45+ cells isolated from the TME.

| external_gene_name | description | TdTOSX+CD45+ logFC | TdTOSX+CD45+ linearFC |
|---|---|---|---|
| Cd163l1 | CD163 molecule-like 1 | 6.482112 | 89.394369 |
| Ltb | lymphotoxin B | 6.217037 | 74.389998 |
| Klrg1 | killer cell lectin-like receptor subfamily G, member 1 | 5.915697 | 60.367363 |
| Foxp3 | forkhead box P3 | 5.882423 | 58.991007 |
| Trbv19 | T cell receptor beta, variable 19 | 5.83901 | 57.242322 |
| Cd5 | CD5 antigen | 5.822965 | 56.609196 |
| Gm4759 | predicted gene 4759 | 5.712238 | 52.426999 |
| Tcrg-C1 | T cell receptor gamma, constant 1 | 5.709921 | 52.342868 |
| Icos | inducible T cell co-stimulator | 5.588856 | 48.129715 |
| Gm19585 | predicted gene, 19585 | 5.577582 | 47.755056 |
| Fasl | Fas ligand (TNF superfamily, member 6) | 5.571314 | 47.548033 |
| Rln3 | relaxin 3 | 5.518062 | 45.824955 |
| Ubash3a | ubiquitin associated and SH3 domain containing, A | 5.432308 | 43.180511 |
| Pcsk1 | proprotein convertase subtilisin/kexin type 1 | 5.376329 | 41.537117 |
| Ccr8 | chemokine (C-C motif) receptor 8 | 5.32423 | 40.06386 |
| Gimap3 | GTPase, IMAP family member 3 | 5.257508 | 38.253185 |
| Slamf1 | signaling lymphocytic activation molecule family member 1 | 5.220641 | 37.288037 |
| Klrb1f | killer cell lectin-like receptor subfamily B member 1F | 5.204067 | 36.86212 |
| Ikzf3 | IKAROS family zinc finger 3 | 5.185332 | 36.386518 |
| Trdv4 | T cell receptor delta variable | 5.179288 | 36.234394 |
| Trat1 | T cell receptor associated transmembrane adaptor 1 | 5.144648 | 35.374755 |
| Pdcd1 | programmed cell death 1 | 5.092017 | 34.107504 |
| Izumo1r | IZUMO1 receptor, JUNO | 5.053635 | 33.212045 |
| Cd3e | CD3 antigen, epsilon polypeptide | 5.042174 | 32.949246 |
| Itk | IL2 inducible T cell kinase | 5.027182 | 32.608632 |
| Ctla4 | cytotoxic T-lymphocyte-associated protein 4 | 4.978043 | 31.516668 |
| Actn2 | actinin alpha 2 | 4.919967 | 30.27316 |
| Trbv1 | T cell receptor beta, variable 1 | 4.891164 | 29.674756 |
| Dkkl1 | dickkopf-like 1 | 4.820628 | 28.258804 |
| Cd300e | CD300E molecule | 4.801355 | 27.883791 |
| Pglyrp2 | peptidoglycan recognition protein 2 | 4.80032 | 27.863797 |
| Cd226 | CD226 antigen | 4.766441 | 27.21709 |
| Themis | thymocyte selection associated | 4.760082 | 27.097397 |
| Dpep3 | dipeptidase 3 | 4.694473 | 25.892693 |
| Cd209a | CD209a antigen | 4.650261 | 25.111233 |
| Cd27 | CD27 antigen | 4.648725 | 25.084512 |
| Gpr55 | G protein-coupled receptor 55 | 4.621113 | 24.608975 |
| Ccr6 | chemokine (C-C motif) receptor 6 | 4.596004 | 24.18439 |
| Gpr174 | G protein-coupled receptor 174 | 4.594683 | 24.16225 |
| A630023P12Rik | RIKEN cDNA A630023P12 gene | 4.586499 | 24.025577 |
| Cd28 | CD28 antigen | 4.564255 | 23.657981 |
| Cd3g | CD3 antigen, gamma polypeptide | 4.547609 | 23.386577 |
| Lta | lymphotoxin A | 4.534293 | 23.171712 |

*Table 2 continued on next page*

*Table 2 continued*

| external_gene_name | description | TdTOSX+CD45+ logFC | TdTOSX+CD45+ linearFC |
|---|---|---|---|
| Ccr3 | chemokine (C-C motif) receptor 3 | 4.526527 | 23.047318 |
| Gzmb | granzyme B | 4.518658 | 22.921957 |
| Lrrc66 | leucine rich repeat containing 66 | 4.47147 | 22.184347 |
| Xirp2 | xin actin-binding repeat containing 2 | 4.462937 | 22.053513 |
| St8sia1 | ST8 alpha-N-acetyl-neuraminide alpha-2,8-sialyltransferase 1 | 4.460924 | 22.022769 |
| Sit1 | suppression inducing transmembrane adaptor 1 | 4.417363 | 21.367754 |
| Tnfrsf4 | tumor necrosis factor receptor superfamily, member 4 | 4.381868 | 20.84844 |

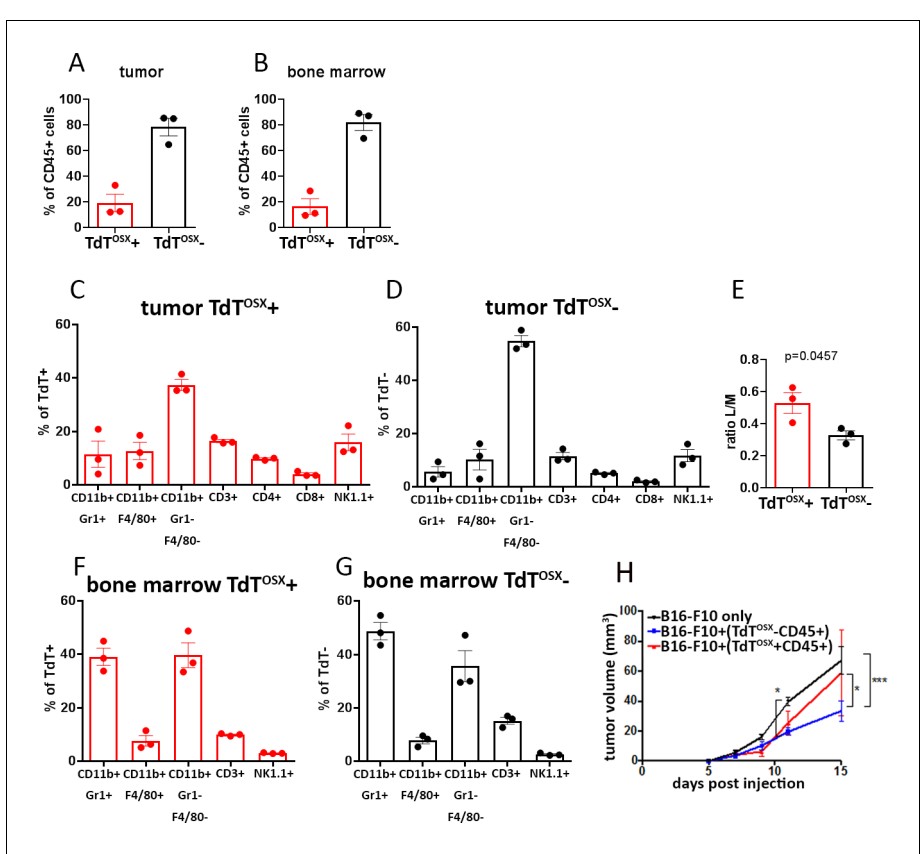

**Figure 6.** Subsets of myeloid and lymphoid cells in the tumor microenvironment and the bone marrow are derived from TdTOSX+ cells. (A–B) Quantification of FACS analysis showing the percentage of TdTOSX+;CD45+ and TdTOSX-;CD45+ populations in the tumor and bone marrow of Osx-cre;TdT mice injected subcutaneously with B16-F10 tumor cells. Data showed as % of total CD45+ cells. (C–D) Quantification of FACS analysis showing the percentage of tumor infiltrating myeloid and lymphoid populations within the TdTOSX+;CD45+ or TdTOSX-;CD45+ subsets, each considered as 100%. (E) Lymphoid over myeloid ratio within the tumor infiltrating TdTOSX+;CD45+ or the TdTOSX-;CD45+ subsets. Statistical analysis was performed by student t-test. (F–G) Quantification of FACS analysis showing the percentage of the bone marrow resident myeloid and lymphoid populations within the TdTOSX+;CD45+ or TdTOSX-;CD45+ subsets, each considered as 100%. n = 3/group. (H) Tumor growth in mice injected with B16-F10 tumor cells alone or together with tumor-derived TdTOSX+;CD45+ or TdTOSX-;CD45+ cells at the ratio 1:5. n = 3–6/group.

The online version of this article includes the following source data and figure supplement(s) for figure 6:

**Source data 1.** Relates to FACS analysis.

**Figure supplement 1.** Gate strategy for immune staining of tumor and bone marrow.

Collectively, these data indicate that *Sp7* marks cells of both mesenchymal and hematopoietic origin to support the development of tumor promoting populations.

## Double positive TdT^OSX+;CD45+ cells derive from TdT^OSX+ HSCs

To determine whether *Sp7* may be activated at early stages of hematopoiesis as a potential explanation for the heterogeneity of TdT^OSX+;CD45+ cells, we performed flow cytometric analysis for hematopoietic stem cell (HSC) markers in the bone marrow of doxy-fed Osx-cre;TdT and TAM-induced Osx-creER^T2;TdT mice bearing B16-F10 tumors subcutaneously. We analyzed the lineage negative (Lin^-) LSK and LK populations (*Figure 7A*) gated based on the expression of Sca1 and c-kit and found 10.9–11.45% TdT^OSX+ LSK and 7.08–8.94% TdT^OSX+ LK in Osx-cre;TdT and Osx-creER^T2; TdT mice, respectively (*Figure 7B and D*). The LSK population was further divided into HSC and multipotent progenitor MPP(1-4) subsets (*Figure 7A* and gate strategy in *Figure 7—figure supplement 1*). Interestingly, TdT^OSX+ cells represented about 30% of MPP1, and between 7–11% of the MPP2-MPP3 and MPP4 subsets in Osx-cre;TdT mice (*Figure 7C*). Similar results were also obtained in Osx-creER^T2;TdT mice (*Figure 7E*). The slight difference in frequency between the two mouse models could depend on the different timing of Cre activation, but both models succeeded to find TdT^OSX+ cells in HSCs.

Importantly, RT-PCR of isolated HSCs from the bone marrow of 7–8 week old wild-type naïve mice showed *Sp7* expression compared to the TdT^OSX- fraction isolated from the tumor mass, used as negative control (*Figure 8A*). Similarly, *Sp7* transcripts were found in the MPP1-4 subsets, and in the common lymphoid progenitors CLP, but not in the common myeloid progenitors CMP

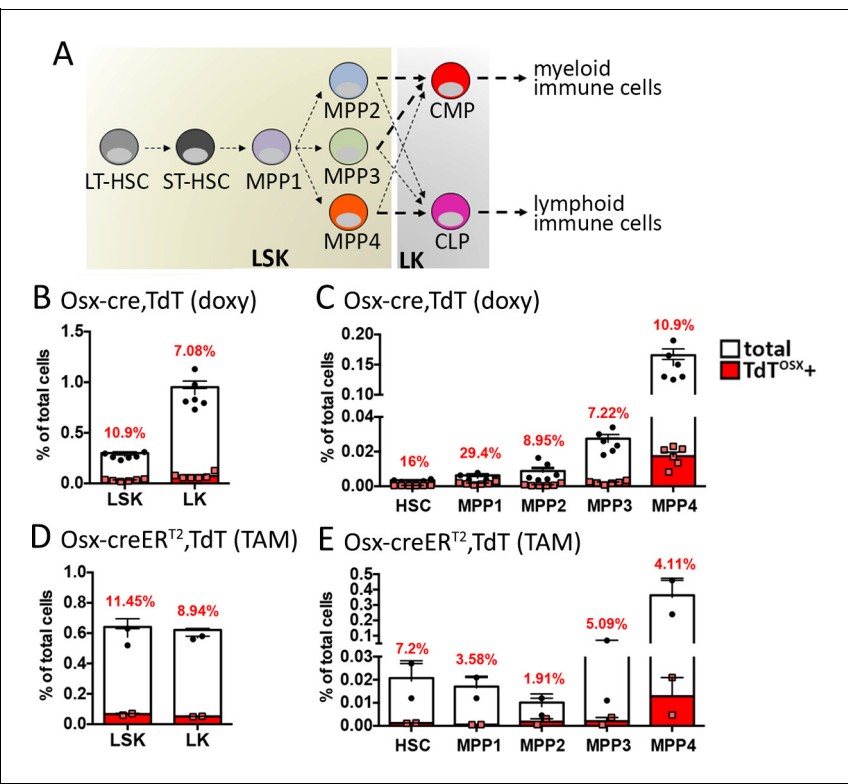

**Figure 7.** TdT^OSX+;CD45+ cells derive from TdT^OSX+ HSCs. (**A**) Schematic of hematopoietic differentiation. (**B–E**) Flow cytometry quantification of TdT^OSX+ cells in the bone marrow of (**B–C**) doxy-fed Osx-cre;TdT mice and (**D–E**) TAM-treated Osx-creER^T2;TdT mice injected subcutaneously at 7 weeks of age with B16-F10 tumors. In (**B and D**) LSK and LK subsets are shown, while in (**C and E**) the LSK population is further divided into HSCs and MPP1-4 subsets. Red numbers represent the average of TdT^OSX+ cells in each specific subset. n = 2–6/group.

The online version of this article includes the following source data and figure supplement(s) for figure 7:

**Source data 1.** Relates to FACS analysis.

**Figure supplement 1.** Gate strategy for HSC immunostaining of bone marrow.

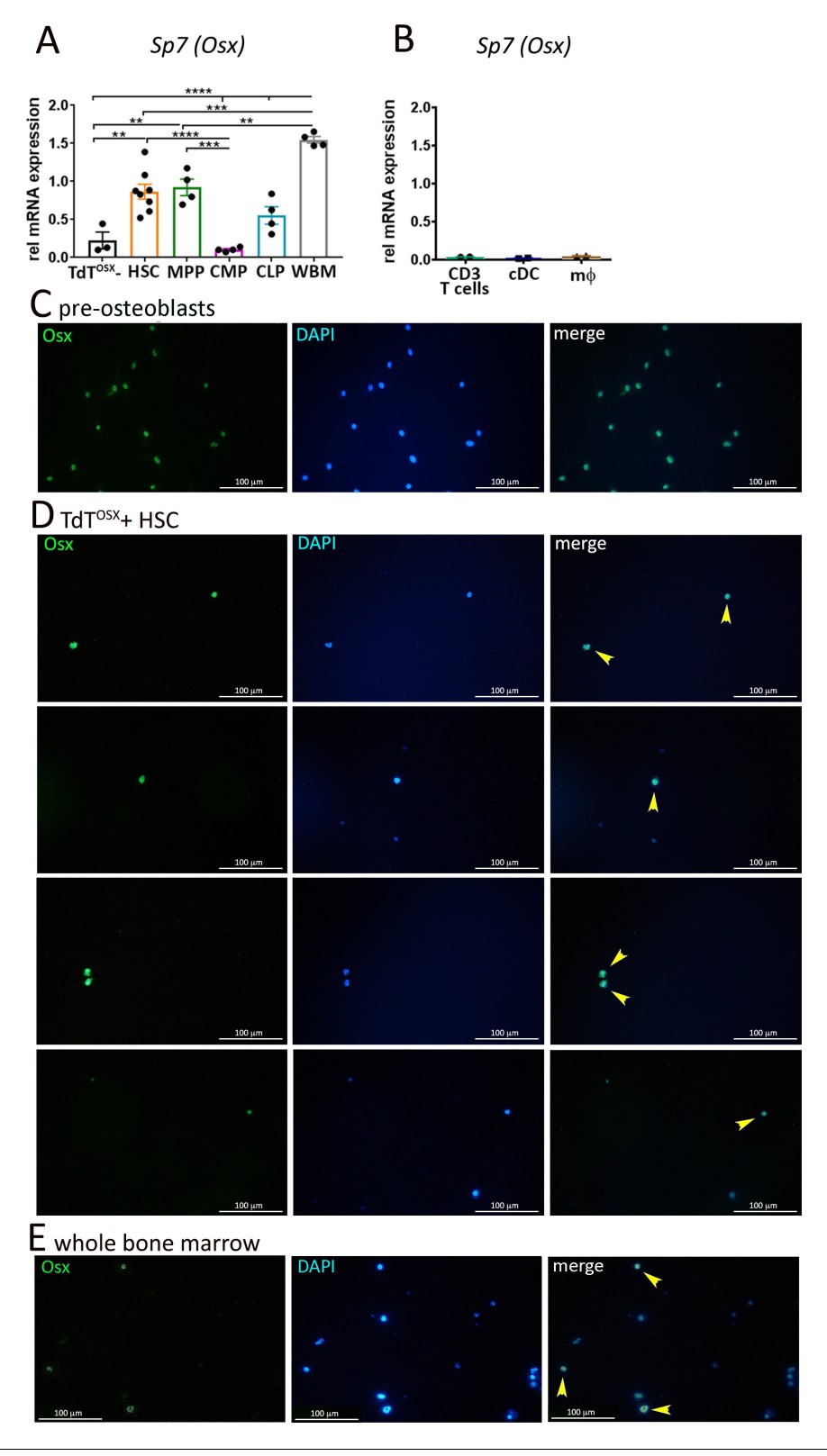

**Figure 8.** *Sp7* transcripts and Osx protein are expressed in a subset of HSCs isolated from bone marrow of naïve mice. (**A**) Real Time PCR analysis comparing sorted hematopoietic stem cells (HSC), multipotent progenitors (MPP), common myeloid progenitors (CMP), common lymphoid progenitors (CLP) and whole bone marrow (WBM) from 7 to 9 week old mice. FACS sorted TdT$^{OSX}$- cells from the tumor of Osx-cre;TdT mice were used as negative control (n = 3–8/group). Statistical analysis was determined by two-way ANOVA followed by Tukey post-hoc test. *p<0.05, **p<0.01, ***p<0.001,
*Figure 8 continued on next page*

*Figure 8 continued*

****p<0.0001. (B) Real Time PCR analysis of isolated mature CD3+ T cells, conventional dendritic cells (cDC) and bone marrow-derived macrophages (mφ) from WT naïve mice. (C–E) Immunofluorescence for Osx in (C) primary osteoblasts differentiated from BMSC cultured for 4 days in osteogenic media, (D) TdT$^{OSX}$+ HSCs sorted from Osx-cre;TdT reporter mice and (E) whole bone marrow cells from WT mice. DAPI is used for nuclear staining. Magnification 200X.

The online version of this article includes the following source data for figure 8:

**Source data 1.** Relates to Real-Time PCR in panels A, B.

(*Figure 8A*). In contrast, RT-PCR analysis of mature immune populations from naïve mice (macrophages, dendritic cells and T cells) failed to detect *Sp7* expression (*Figure 8B*). Finally, we performed immunostaining for Osx to confirm protein expression on a single cell level. To enrich for Osx+ cells, we FACS sorted TdT$^{OSX}$+ HSCs from the bone marrow of Osx-cre;TdT reporter mice after enrichment for c-kit+ cells. Sorted cells were plated on serum-coated coverslips for two days before immunostaining for Osx and with DAPI for nuclear localization. We confirmed Osx localization in the nuclei of the majority of TdT$^{OSX}$+ HSCs (*Figure 8D*). As positive control for Osx staining, we used bone marrow stromal cells isolated from WT mice cultured for 4 days in osteogenic medium (*Figure 8C*). Whole bone marrow was used as additional control confirming that only a small number of cells expressed Osx while the majority was negative (*Figure 8E*). Thus, for the first time we show that subsets of HSCs express the transcription factor *Sp7*, giving rise to tumor infiltrating immune populations.

## Discussion

Numerous studies have established the importance of the TME for primary tumor growth and metastatic dissemination (*Goubran et al., 2014*). In skeletal metastases, crosstalk between bone residing cells, osteoblasts and osteoclasts, and tumor cells drives tumor growth (*Weilbaecher et al., 2011*). Recent studies also suggest that osteoblasts and their secretory products stimulate the expansion of bone marrow-derived myeloid populations, which in turn escape the bone marrow and reach distant sites to support tumor growth by inhibiting anti-tumor immune responses (*D'Amico et al., 2016*; *Engblom et al., 2017*). Our study demonstrates that a transcription factor required for osteoblast differentiation, Osterix, marks pro-tumorigenic stromal cells infiltrating extra-skeletal tumors. A possible role of Osx in tumorigenesis is emerging from human studies as well showing that presence of *Sp7* in the tumor is associated with poor patient survival (*Yao et al., 2019*). Our own analysis of published gene microarray data of breast cancer stroma (*Finak et al., 2008*), reveals higher *Sp7* expression in tumor stroma relative to healthy tissue from the same subject, corroborating the notion that Osx+ stromal cells regulate tumor growth. Therefore, Osx function may well extend beyond its known role in bone development and homeostasis.

Osterix has been primarily studied in the context of osteoblast differentiation and regulation of bone mass. *Sp7* deficient mice die within 1 hr of birth with a complete absence of intramembranous and endochondral bone formation (*Nakashima et al., 2002*; *Baek et al., 2009*). During embryogenesis and perinatally, *Sp7* is expressed in MSCs in the bone marrow, and it is necessary for the full osteogenic program. In adult mice, *Sp7* is mainly confined to committed osteoblasts and osteocytes. There is increasing evidence of *Sp7* expression in cells residing outside the skeleton, including synovial fibroblasts (*Miura et al., 2019*), dental pulp (*Monterubbianesi et al., 2019*), olfactory glomerular cells, gastric and intestinal epithelium (*Chen et al., 2014*), and kidney (*Strecker et al., 2013*). Here we demonstrate that *Sp7* is also present in a CAF subset, and in hematopoietic precursors, thus challenging the dogma that *Sp7* is exclusively expressed by mesenchymal cells. Our cell tracking studies using the repressible Tet-OFF Osx-cre;TdT reporter and the tamoxifen-inducible Osx-creER$^{T2}$;TdT model demonstrate the presence of TdT$^{OSX}$+;CD45+ cells in the bone marrow of adult mice. At least two other independent studies have also shown abundant and persistent Osx+ cells in the bone marrow when the reporter gene is activated either embryonically or neonatally (*Liu et al., 2013*; *Mizoguchi et al., 2014*), and at 3 weeks of age (*Strecker et al., 2013*), but no signal in the bone marrow when the Osx-cre is activated after 8 weeks of age (*Mizoguchi et al., 2014*). Notably, most of the studies with Osx-cre reporter mice have excluded the hematopoietic marker CD45 in the analysis of the Osx+ populations, and no report to date has shown *Sp7* expression in the

hematopoietic compartment. Importantly, we also report that bone marrow residing TdT^OSX+ populations expand during tumor progression. Such result is highly unlikely linked to systemic factors released from extra-skeletal tumors enhancing the Tet-transactivator in the Tet-OFF;Osx-cre;TdT mice. In fact, we observed increased TdT^OSX+ cells (CD45- and CD45+ subsets) also in the bone marrow of tumor bearing Osx-creER^T2;TdT animals. Our findings are instead consistent with previous observations reporting increased numbers of bone marrow residing MSCs and osteoblast precursors in mice and patients with lung adenocarcinomas (*Engblom et al., 2017*), and with the reprogramming of hematopoiesis towards increased myelopoiesis during tumor progression (*Capietto et al., 2013*; *Meyer et al., 2018*).

Here, we report for the first time that a subset of hematopoietic lineage cells in the bone marrow and at tumor site derives from an Osx+ progenitor, which is present embryonically and persists until 7–8 weeks of age. Importantly, we detected *Sp7* transcripts and Osx nuclear localization in a subset of HSCs. These results support the observation that TdT^OSX+;CD45+ cells are very heterogeneous and express monocyte, macrophage, granulocyte, T cell and NK cell markers. The heterogeneity of the TdT^OSX+;CD45+ populations is most likely derived from expression of *Sp7* in HSC and MPP subsets. These precursors can give rise to the lineage committed common lymphoid (CLP) and myeloid (CMP) progenitors that differentiate into all the mature immune cells (*Pietras et al., 2015*). *Sp7* transcripts are also detected in CLP but not in CMP or mature immune populations, indicating that Osx is activated early during hematopoiesis, with higher expression in cells committed to the lymphoid lineage. Despite expressing both myeloid and lymphoid markers, we noted increased in lymphoid over myeloid ratio in the TdT^OSX+;CD45+ tumor infiltrating cells relative to the TdT^OSX-;CD45+ cells. TdT^OSX+;CD45+ subset is also enriched in genes expressed by immunosuppressive or exhausted lymphocytes. Such result is consistent with the functional tumor co-injection studies showing that TdT^OSX-;CD45+ cells reduce tumor growth while the TdT^OSX+;CD45+ subset does not.

Co-injection of tumor cells with cells sorted from the primary tumor has been used to validate the pro-tumorigenic roles of CAFs and certain immune suppressive CD45+ populations. Here, we show that tumor-derived TdT^OSX+ cells, comprising both CD45 positive and negative subsets, increase tumor growth compared to tumor cells injected alone. By contrast, co-injection of bone marrow-derived TdT^OSX+ cells with the tumor cells does not exert any pro-tumor effect suggesting that bone marrow-derived TdT^OSX+ cells are functionally distinct from tumor-derived TdT^OSX+ cells. Hence, if tumor-derived TdT^OSX+ cells originate from the bone marrow in response to an incipient tumor, as it would be anticipated by their increase in the circulation of tumor-bearing mice, they must undergo a conditioning process within the TME to acquire a pro-tumorigenic function. Because Osx regulates expression of proteins required for the generation of the bone tissue, it is likely that a main role of Osx+ mesenchymal cells (TdT^OSX+;CD45-) in the TME is to produce and remodel extracellular matrix. Fibroblasts with an osteoblast signature involved in matrix production have been recently identified by single cell RNA sequencing in rheumatoid arthritis (*Croft et al., 2019*). ECM accumulation is quite frequent within the TME, causing in more severe cases an intense fibrotic response, or desmoplasia, and tumor stiffening (*Gkretsi and Stylianopoulos, 2018*). Stiffening is not only required for a primary tumor to displace the host tissue and grow in size, but also contributes to cell–ECM interactions and can promote cancer cell invasion to surrounding tissues (*Gkretsi and Stylianopoulos, 2018*). As noted earlier, upregulation of *Sp7* in breast cancer cells is associated with increased invasion and bone metastasis, and this may occur by upregulation of ECM modifying metalloproteinases, MMP9 and MMP13, and other factors that increase vascularization, and affect bone cell function (*Yao et al., 2019*).

Since *Sp7* is not expressed in mature immune populations, it is difficult to envision a role for Osx in modulating anti-tumor immune responses. Co-injection experiments indicate that TdT^OSX+;CD45+ cells have better pro-tumorigenic function than TdT^OSX–;CD45+ cells. However, only the co-injection of the bulk TdT^OSX+ cells increases tumor growth over the B16-F10 tumor cells alone, suggesting that the Osx+ mesenchymal population (TdT^OSX+;CD45-) is responsible for enhancing tumor growth. This finding is intriguing since the TdT^OSX+;CD45- mesenchymal cells represent only 5% of the total tumor infiltrating TdT^OSX+ populations. One could speculate that TdT^OSX+;CD45- cells represent a subset of highly pro-tumorigenic CAFs, which can support tumor progression even when present in very limited numbers. Another possibility is that the TdT^OSX+;CD45- mesenchymal cells might be required to create and maintain an immune suppressive environment where the CD45+ populations fail to exert anti-tumor effects.

In conclusion, we demonstrate that *Sp7* is expressed in a subset of tumor infiltrating mesenchymal cells with CAF and osteogenic cell features. Surprisingly, *Sp7* expression is also found in hematopoietic precursors, and marks tumor infiltrating immune populations enriched in immune suppressive markers. Considering the emerging data that *Sp7* expression in the tumor cells is linked to tumor progression, our results emphasize the importance of Osx in the TME and the need to evaluate the prognostic value of stromal *Sp7* expression in the patients.

# Materials and methods

## Key resources table

| Reagent type (species) or resource | Designation | Source or reference | Identifiers | Additional information |
|---|---|---|---|---|
| Gene (*Mus musculus*) | Sp7(Osx) | NCBI Gene | RRID:MGI:2153568 | NCBI ID:170574 |
| Genetic reagent (*Mus musculus*) | Tg(Sp7-tTA,tetO-EGFP /cre)1Amc/J (Osx-cre) | The Jackson Laboratories | Cat#006361 RRID:MGI:3689350 | C57Bl/6 |
| Genetic reagent (*Mus musculus*) | B6.Cg-Gt(ROSA)26Sortm9 (CAG-tdTomato)Hze/J (TdT) | The Jackson Laboratories | RRID:MGI3813511 | C57Bl/6 |
| Genetic reagent (*Mus musculus*) | Sp7-creERT2/Ai9tdTomato (Osx-creERT2;TdT) | PMID:31768488 | Cat#007909 RRID:MGI:4829803 | Prof. Silva MJ (Washington University in St Louis) C57Bl/6 |
| Cell line (*Mus musculus*, C57Bl/6) | B16-F10 melanoma cell line | ATCC | RRID:CRL-6475-LUC2 | C57Bl/6 |
| Cell line (*Mus musculus*) | PyMT breast cancer cell line | PMID:27216180 | | Prof. Weilbaecher KN (Washington University in St Louis) C57Bl/6 |
| Cell line (*Mus musculus*) | Immortalized Cancer Associated Fibroblast cell line | PMID:27264173 | | Prof Longmore GD (Washington University in St Louis) |
| Antibody | Rat anti-mouse CD45-APCeFluor780 (clone 30-F11) | eBioscience | Cat #47-0451-82 RRID:AB_1548781 | FACS (1:400) |
| Antibody | Rat anti-mouse CD45-PE Cy7 (clone 30-F11) | eBioscience | Cat #25-0451-82 RRID:AB_2734986 | FACS (1:400) |
| Antibody | Rat anti-mouse CD11b-AlexaFluor700 (clone M1/70) | eBioscience | Cat#56-0112-82 RRID:AB_657585 | FACS (1:400) |
| Antibody | Rat anti-mouse Gr1(Ly6G)-FITC (clone RB6-8C5) | Miltenyi Biotec | Cat#130-102-837 RRID:AB_2659858 | FACS (1:100) |
| Antibody | Rat anti-mouse F4/80 PerCP Cy5.5 (clone BM8) | Biolegend | Cat#123126 RRID:AB_10802654 | FACS (1:400) |
| Antibody | Hamster anti-mouse CD3e-PE (clone 145–2 C11) | eBioscience | Cat#A14714 RRID:AB_2534230 | FACS (1:200) |
| Antibody | Rat anti-mouse CD4-APC (clone RM4-5) | BD Biosciences | Cat#553051 RRID:AB_398528 | FACS (1:200) |
| Antibody | Rat anti-mouse CD8a-FITC (clone 53–6.7) | BD Biosciences | Cat#561966 RRID:AB_10896291 | FACS (1:200) |
| Antibody | Mouse anti-mouse NK1.1 (clone PK136) | Biolegend | Cat#108722 RRID:AB_2132712 | FACS (1:200) |
| Antibody | Rat anti-mouse CD45R (B220)-Biotin (clone RA3-6B2) | eBioscience | Cat#13-0452-82 RRID:AB_466449 | FACS (0.4 ul/sample) |
| Antibody | Hamster anti-mouse CD3e-Biotin (clone 145–2 C11) | eBioscience | Cat#13-0031-82 RRID:AB_466319 | FACS (0.4 ul/sample) |

*Continued on next page*

*Continued*

| Reagent type (species) or resource | Designation | Source or reference | Identifiers | Additional information |
|---|---|---|---|---|
| Antibody | Rat anti-mouse Ter119 (clone TER119) | eBioscience | Cat# 13-5921-82 RRID:AB_466797 | FACS (0.4 ul/sample) |
| Antibody | Rat anti-mouse Ly-6G/Ly-6C Biotin (clone RB6-8C5), | eBioscience | Cat#13-5931-82 RRID:AB_466800 | FACS (0.2 ul/sample) |
| Antibody | Rat anti-mouse CD41Biotin (clone MWReg30) | eBioscience | Cat#13-0411-82 RRID:AB_763484 | FACS (1 ul/sample) |
| Antibody | Streptavidin-BV510 | BD Biosciences | Cat#563261 | FACS (0.5 ul/sample) |
| Antibody | Rat anti-mouse Sca-1 BV711 (clone D7) | BD Biosciences | Cat#563992 RRID:AB_2738529 | FACS (0.25 ul/sample) |
| Antibody | Rat anti-mouse c-kit(CD117)-APCeFluor780 (clone ACK2) | eBioscience | Cat#47-1172-82 RRID:AB_1582226 | FACS (1 ul/sample) |
| Antibody | Rat anti-mouse CD16/32-BUV395 (clone 2.4G2) | BD Biosciences | Cat#740217 RRID:AB_2739965 | FACS (0.5 ul/sample) |
| Antibody | Rat anti-mouse CD34-FITC (clone RAM34) | eBioscience | Cat#11-0341-82 RRID:AB_465021 | FACS (4 ul/sample) |
| Antibody | Rat anti-mouse CD150 (SLAM)-BV421 (clone TC15-12F12.2) | Biolegend | Cat#115943 RRID:AB_2650881 | FACS (1 ul/sample) |
| Antibody | Hamster anti-mouse CD48-PE Cy7 (clone HM48-1) | eBioscience | Cat#25-0481-80 RRID:AB_1724087 | FACS (0.15 ul/sample) |
| Antibody | Rat anti-mouse CD135(Flt3)-APC (clone A2F10) | eBioscience | Cat#17-1351-82 RRID:AB_10717261 | FACS (2 ul/sample) |
| Antibody | Rat anti-mouse CD45 (clone 30-F11) | Invitrogen | Cat#14-0451-82 RRID:AB_467251 | IF, IHC (1:200) |
| Antibody | Rabbit anti-mouse RFP/TdT | Rockland | Cat#600-401-379 RRID:AB_2209751 | IF,IHC (1:500) |
| Antibody | Goat-anti-Rat IgG-Biotin | Thermo Fisher | Cat#31830 RRID:AB_228355 | IF, IHC (1:500) |
| Antibody | Streptavidin-HRP | Bio-Rad | Cat#STAR5B | IF, IHC (1:500) |
| Antibody | Goat-anti-rabbit Alexa-fluor647 | Thermo Fisher | Cat#A21244 RRID:AB_2535812 | IF (4 ug/ml) |
| Antibody | Rabbit anti-mouse Sp7/Osx | Abcam | Cat#ab227820 | IF (1:500) |
| Antibody | Goat anti-rabbit AlexaFluor488 | Abcam | Cat#ab150077 RRID:AB_2630356 | IF (1:1000) |
| Sequence-based reagent | Cyclophilin | This paper | PCR primers | 5'-AGC ATA CAG GTC CTG GCA TC-3' and 5'-TTC ACC TTC CCA AAG ACC AC-3' |
| Sequence-based reagent | Sp7(Osx) | This paper | PCR primers | 5'-AAG GGT GGG TAG TCA TTT GCA-3' and 5'-CCC TTC TCA AGC ACC AAT GG-3' |
| Sequence-based reagent | S100A4(Fsp-1) | This paper | PCR primers | 5'-TGA GCA ACT TGG ACA GCA ACA-3' and 5'-TTC CGG GGT TCC TTA TCT GGG-3' |
| Sequence-based reagent | Acta2(α-SMA) | This paper | PCR primers | 5'-GTC CCA GAC ATC AGG GAG TAA-3' and 5'-TCG GAT ACT TCA GCG TCA GGA-3' |
| Sequence-based reagent | Col1a1 | This paper | PCR primers | 5'-GGC CTT GGA GGA AAC TTT GC-3' and 5'-GGG ACC CAT TGG ACC TGA AC-3' |

*Continued on next page*

*Continued*

| Reagent type (species) or resource | Designation | Source or reference | Identifiers | Additional information |
|---|---|---|---|---|
| Sequence-based reagent | Bglap(Ocn) | This paper | PCR primers | 5'-GGA CTG AGG CTC TGT GAG GT-3' and 5'-CAG ACA CCA TGA GGA CCA TC-3' |
| Sequence-based reagent | Runx2 | This paper | PCR primers | 5'-GTT ATG AAA AAC CAA GTA GCC AGG-3' and 5'-GTA ATC TGA CTC TGT CCT TGT GGA-3' |
| Sequence-based reagent | BSP | This paper | PCR primers | 5'-AGG ACT AGG GGT CAA ACA C-3' and 5'-AGT AGC GTG GCC GGT ACT TA-3' |
| Sequence-based reagent | TNAP | This paper | PCR primers | 5'-GGG GAC ATG CAG TAT GAG TT-3' and 5'-GGC CTG GTA GTT GTT GTG AG-3' |
| Commercial assay or kit | TSA FITC System | Perkin Elmer | Cat#NEL701A001KT | |
| Commercial assay or kit | BOND Intense R detection kit | Leica Biosystem | Cat#DS9263 | |
| Commercial assay or kit | BOND Polymer Refine Red detection kit | Leica Biosystem | Cat#DS9390 | |
| Commercial assay or kit | Avidin/Biotin Blocking kit | Vector Labs | Cat#SP2001 RRID:AB_2336231 | |
| Software, algorithm | STAR | STAR | RRID:SCR_015899 | Version 2.0.4b |
| Software, algorithm | Subread | Subread | RRID:SCR_009803 | Version 1.4.5 |
| Software, algorithm | R | R | RRID:SCR_001905 | Version 3.4.1 |
| Software, algorithm | EdgeR | EdgeR | RRID:SCR_012802 | Version 3.20.2 |
| Software, algorithm | LIMMA | LIMMA | RRID:SCR_010943 | Version 3.34.4 |
| Software, algorithm | GAGE | GAGE | RRID:SCR_017067 | Version 2.28.0 |
| Other | VECTASHIELD Mounting Medium with DAPI | Vector Labs | Cat#H1200 RRID:AB_2336790 | one drop |
| Other | ProLong Gold with DAPI | Invitrogen | Cat#P10144 | one drop |

## Tumor cell lines, CAF isolation and generation of primary cell cultures

B16-F10 (C57BL/6 mouse melanoma cell line, ATCC # CRL-6475-LUC2), PyMT and PyMT-BO1-GFP (*Su et al., 2016*) (C57BL/6 mouse breast cancer cell lines), kindly provided by Prof. Katherine N Weil-baecher (Washington University in St. Louis), were cultured at 37°C in complete media (DMEM supplemented with 2 mM l-glutamine, 100 µg/ml streptomycin, 100 IU/ml penicillin, and 1 mM sodium pyruvate) containing 10% FBS. Immortalized CAF cell line was kindly provided by Prof. Gregory D Longmore (Washington University in St. Louis). All the cell lines used have been tested for myco-plasma and were mycoplasma free. CAFs were isolated from primary tumors as described in *Corsa et al., 2016*. Tumors were minced, digested and plated as single cell suspension on a tissue plastic dish for 30 min to separate the adherent fraction, composed of CAFs, from the cells in suspension, comprising of tumor cells, tumor infiltrating immune populations and endothelial cells, among few others. CAFs were then left 24 hr in culture and RNA was extracted afterwards.

Bone marrow stromal cells (BMSCs) were cultured in complete alpha-MEM (without ascorbic acid) containing 10% FBS and differentiated for 4 days in osteogenic medium (complete alpha-MEM supplemented with 50 µg/ml ascorbic acid and 10 µM beta-glycerophosphate) to obtain pre-osteoblasts.

Sorted hematopoietic stem cells (HSCs) were plated overnight on a glass coverslip coated with serum in the presence of StemPRO (Gibco) media and then fixed and used for immunostaining.

## Mouse strains and tumor models

Animals were housed in a pathogen-free animal facility at Washington University (St. Louis, MO, USA). Age and sex-matched animals were used in all experiments according to IACUC guidelines. *B6.Cg-Tg(Sp7-tTA,tetO-EGFP/cre)1Amc/J* (catalog #006361; The Jackson laboratory, ME USA) mice, which carry a tetracycline-responsive Osx promoter driving Cre (Osx-cre) were mated with reporter mice *B6.Cg-Gt(ROSA)26Sor$^{Tm9(CAG-tdTomato)Hze}$/J* (TdT) (catalog #007909; The Jackson Laboratory), to generate Osx-cre;TdT mice. To suppress Cre expression, 200ppm doxycyline (doxy) was added to the chow (Test Diet #1816332–203, Purina, MO USA) and fed to some groups of mice until weaning (P28); pups were switched to standard rodent chow at weaning. *Sp7-creER$^{T2}$/Rosa26$^{<fs-TdTomato>}$* mice were kindly provided by Prof. Matthew D Silva (Washington University in St. Louis). Tamoxifen (#T5648, Millipore Sigma, MO USA) was dissolved in corn oil and injected at 100 mg/kg intra-peritoneally. Wild-Type (WT) C57BL/6 mice were purchased from The Jackson Laboratory.

Subcutaneous (sq) injections were performed using $10^5$ B16-F10 tumor cells suspended in 100 µl of sterile PBS and Matrigel Matrix (Corning #354234), while $10^5$ PyMT were suspended in 50 µl of sterile PBS and Matrigel Matrix and injected into the MFP. Adoptive transfer and co-injection with tumor cells were performed using a 5:1 ratio of TdT$^{OSX+}$:B16-F10 tumor cells. The number of cells injected in each experiment varied depending upon the number of cells obtained from sorting in each experiment (hence, the differences in tumor size among the different experiments), but within ($2.5×10^5$:$5×10^4$) and ($5×10^5$:$1×10^5$) cells. B16-F10 cells only were injected as control using the same number of tumor cells co-injected with TdT$^{OSX}$+ cells. Tumors were monitored by caliper measurements and mice were sacrificed between day 14 and 16 post tumor inoculation. Tumor volume was calculated according to the formula: 0.5236*length*(width [*Kalluri and Zeisberg, 2006*]).

## Flow cytometry and sorting

Single cell suspensions were prepared from fresh bone marrow and tumors upon sacrifice. Bone marrow cells were obtained from femurs and tibias by centrifugation, while tumors were minced and digested with 3.0 mg/ml collagenase A (Roche, Basel Switzerland) and 50 U/ml DNase I (Millipore Sigma) in serum-free media for 45 min at 37°C. Cells were filtered through 70 µm strainers, red blood cells (RBC) were then removed with RBC lysis buffer (Millipore Sigma), washed twice in PBS and stained in FACS buffer (0.5% BSA, 2 mM EDTA, 0.01%NaN$_3$) with the following antibodies: CD45 (clone 30-F11, eBioscience), CD11b (clone M1/70, eBioscience), Ly6G (Gr-1) (clone RB6-8C5, eBioscience), F4/80 (BM8, Biolegend), CD3 (145–2 C11, eBioscience), CD4 (RM4-5, BD Biosciences), CD8 (clone 53–6.7, BD Biosciences), NK1.1 (clone PK136, Biolegend). Once stained, cells were sorted for the adoptive transfer, RT-PCR and RNAseq studies or were fixed for flow cytometry (gate strategy in *Figure 7—figure supplement 1*) in BD Cytofix fixation buffer (BD Biosciences, 554655). Samples were acquired using the BD LSR-Fortessa cytometer and analyzed with FlowJo version 9.3.2.

Hematopoietic stem cell subsets were analyzed from fresh bone marrow isolated from femurs, tibias and hip bones crushed with mortar and pestle in 3 ml of PBS, filtered in a 70µm-nylon strainer to remove bone fragments and lysed of red blood cells (RBC). 10 million cells were stained for each sample using the following antibodies: CD3, Gr1, B220 (clone RA3-6B2, eBioscience), Ter119 (clone TER119, e Bioscience), CD41 (clone eBioMWReg30, eBioscience) for lineage negative gating, Sca-1 (clone D7, BD Biosciences), CD117/c-kit (clone ACK2, eBioscience), CD16/32 (clone 2.4G2, BD Biosciences), CD34 (clone RAM34, eBioscience), CD150/SLAM (clone TC15-12F12.2, Biolegend), CD48 (clone HM48-1, eBioscience), CD135/Flt3 (clone A2F10, eBioscience), gate strategy in *Figure 7—figure supplement 1*. Once stained, cells were fixed in BD Cytofix fixation buffer, acquired using the Bio-Rad ZE5 (YETI) cytometer and analyzed with FlowJo version 10.4.1.

Isolation of hematopoietic progenitor subsets for Real Time analysis was obtained by FACS sorting using the following markers: HSCs (Lineage- c-Kit+Sca-1+CD48-CD150+), MPPs (Lineage- c-Kit+Sca-1+CD48-CD150-), CLPs (Lineage- c-Kit+Sca-1+IL7rα+Flt3+Ly6D-) and CMPs (Lineage- c-Kit+Sca-1-CD34+CD16/32). For IF, TdTomato was also used to sort Osx+ HSCs.

Isolation of hematopoietic stem cells for Osx immunostaining was performed as following: whole bone marrow was isolated from femurs, tibias and iliac crests and incubated with mouse CD117-conjugated microbeads (Miltenyi Biotec #130-091-224). CD117-enrichment was then carried out using the AutoMACS Pro Seperator (Miltenyi Biotec). Post CD117 enrichment, the positive cell fraction

was stained to sort phenotypically defined TdTomato+ HSCs (Lineage- c-Kit+Sca-1+CD48-CD150+) by FACS.

## Histology

Tumors and bones (femur and tibia) were fixed in 4% PFA over-night (ON) at 4°C. The fixed bones were partially decalcified in 14% EDTA for 3 days, washed in a sucrose gradient (1 hr in 10% sucrose, 1 hr in 20% sucrose, ON in 30% sucrose) before snap-freezing them in OCT embedding medium. Tumors were directly put in sucrose gradient and then embedded in OCT. Frozen sections were cut at 5 µm thickness and kept at −20°C until analysis.

Thawed sections were washed in PBS and mounted with VECTASHIELD Mounting Medium containing DAPI (Vector Laboratories, CA USA).

Images were taken using the Leica DMi8 Confocal Microscopy at the Musculoskeletal Research Center (Washington University in St Louis), magnification 200X.

## Immunohistochemical staining

Tissues were fixed in 10% neutral-buffered formalin for 18 hr, embedded in paraffin after graded-ethanol dehydration, and sectioned into 6 µm sections using a microtome. Automated staining was carried out on the BondRxm (Leica Biosystems). Following dewaxing and citrate-buffered antigen retrieval, sections were stained with the primary antibody for 1 hr at RT. Sequential chromogenic detection was performed with the Bond IntenseR (Rat primary) and Bond Polymer Refine Red (Rabbit primary) detection kits (Leica Biosystems). The primary antibodies used were rat anti-CD45 (Invitrogen clone 30-F11, #14-0451-82) followed by rabbit anti-RFP/TdT (Rockland, # 600-401-379). Stained sections were dehydrated in graduated ethanol and xylene washes then mounted with xylene-based Cytoseal (Thermo Fisher).

Stained slides were imaged at 20X magnification with the Zeiss Axioscan slide scanner. For image analysis, Halo v3 (Indica Labs) was used to deconvolve dual stained images into single channels and pseudo colored with blue representing nuclei (hematoxylin), green representing CD45+ cells (DAB stained), and red representing Osx+ cells (RFP stained).

## Immunohistochemestry and immunofluorescence staining

For colocalization of TdT+ cells and CD45 in B16-F10 soft tissue tumors, subcutaneous tumors were dissected and fixed in 10% neutral-buffered formalin for 18 hr, embedded in paraffin after graded-ethanol dehydration, and sectioned into 6 µm sections using a microtome. Sections were dewaxed in xylene and then hydrated through graded ethanol washes. Endogenous peroxidases were quenched by incubating in hydrogen peroxide (1% in PBS) for 10 min at RT. Antigen retrieval was then performed by microwave treatment in a citrate buffer for 15 min. Sections were washed in TBS-T (1X TBS with 0.05%Tween-20) followed by blocking for 30 min at RT in blocking buffer (5% goat serum, 2.5% BSA in TBS). Slides were then blocked using the Avidin/Biotin Blocking Kit (Vector Labs). For staining, slides were incubated overnight in a humidified chamber at 4°C with 1:200 rat anti-CD45 (Invitrogen clone 30-F11, #14-0451-82) and 1:400 rabbit anti-RFP (Rockland, #600-401-379) antibodies diluted in 50% blocking buffer. After primary staining, slides were washed in TBS-T and then incubated with HRP-conjugated anti-rat IgG secondary (1:500 in 50% blocking buffer) for 30 min at RT. For fluorescent detection of CD45, slides were washed and then incubated in 1:50 FITC Tyramide reagent (PerkinElmer, # NEL701A001KT) for 8 min at RT. For fluorescent detection of RFP, slides were washed and incubated in AlexaFluor 594-conjugated anti-rabbit IgG secondary (1:500 in 50% blocking buffer) for 30 min at RT. Stained slides were washed and mounted using Pro-Long Gold with DAPI (Invitrogen).

### Osx staining

HSCs, BMSC or whole bone marrow cells were plated overnight on coverslips coated with FBS for 30 min at RT then cells. Cells were fixed in 4% PFA for 15 min, washed and permeabilized with PBS/0.3%TritonX-100 for 3 min. Blocking was performed with 5% normal serum in permeabilization buffer for 1 hr at RT. Anti-Osx antibody (Abcam, #ab227820) was dissolved 1:500 in a buffer containing PBS/0.3%TritonX-100/1%BSA and incubated ON at 4°C. For fluorescent detection, coverslips were washed and incubated in AlexaFluor 488-conjugated anti-rabbit secondary antibody (1:1000) in PBS/

0.3%TritonX-100/1%BSA for 1 hr at RT. Stained slides were washed and mounted using VECTA-SHIELD Mounting Medium containing DAPI (Vector Laboratories, CA USA). Fluorescent signals were captured by using a Nikon Eclipse 80i microscope and a Nikon DS-Qi1MC camera (Nikon, CO, USA).

## Real Time PCR

Total RNA was extracted with TRIzol (Invitrogen, CA USA) and quantified on a ND-1000 spectrophotometer (NanoDrop Technologies). The cDNA was synthesized with 1 µg RNA using High Capacity cDNA Reverse Transcription Kit (#4368814, Applied Biosystems, CA USA).

For purified hematopoietic precursors (HSC, MPP, CLP and CMP) total RNA was extracted using the NucleoSpin RNA XS kit (Macherey-Nagel #740902) and reverse transcribed with the SuperScript VILO kit (Invitrogen #11754–050).

The amount of each gene was determined using Power SYBR Green mix on 7300 Real-Time PCR System (Applied Biosystems). Cyclophilin mRNA was used as housekeeping control. Specific primers for mice were as follows: *Cyclophilin*, 5'-AGC ATA CAG GTC CTG GCA TC-3' and 5'-TTC ACC TTC CCA AAG ACC AC-3'; *Osterix*, 5'-AAG GGT GGG TAG TCA TTT GCA-3' and 5'-CCC TTC TCA AGC ACC AAT GG-3'; *S100a4 (Fsp1)*, 5'-TGA GCA ACT TGG ACA GCA ACA-3' and 5'-TTC CGG GGT TCC TTA TCT GGG-3'; *Acta2 (a-SMA)*, 5'-GTC CCA GAC ATC AGG GAG TAA-3' and 5'-TCG GAT ACT TCA GCG TCA GGA-3'; *Col1a1*, 5'-GGC CTT GGA GGA AAC TTT GC-3' and 5'-GGG ACC CAT TGG ACC TGA AC-3'; *Bglap (Osteocalcin)*, 5'-GGA CTG AGG CTC TGT GAG GT-3' and 5'-CAG ACA CCA TGA GGA CCA TC-3'; *Runx2*, 5'-GTT ATG AAA AAC CAA GTA GCC AGG-3' and 5'-GTA ATC TGA CTC TGT CCT TGT GGA-3'; *Ibsp (Bsp)*, 5'-AGG ACT AGG GGT CAA ACA C-3' and 5'-AGT AGC GTG GCC GGT ACT TA-3'; *Alpl (Tnap)*, 5'-GGG GAC ATG CAG TAT GAG TT-3' and 5'-GGC CTG GTA GTT GTT GTG AG-3'. The relative quantification in gene expression was determined using the $2^{-\Delta\Delta Ct}$ method.

## RNA sequencing and analysis

Single cell suspensions were prepared from fresh tumor upon sacrifice. Tumors were minced and digested as described above. Tumor stromal cells were separated from the GFP+ tumor cells and sorted based on the expression, or lack of thereof, of TdT and CD45 markers.

Library preparation was performed with 10 ng of total RNA and integrity was determined using an Agilent bioanalyzer. ds-cDNA was prepared using the SMARTer Ultra Low RNA kit for Illumina Sequencing (Clontech) per manufacturer's protocol. The cDNA was fragmented using a Covaris E220 sonicator using peak incident power 18, duty factor 20%, cycles/burst 50, for a 120 s. cDNA was blunt ended, had an A base added to the 3' ends, and then had Illumina sequencing adapters ligated to the ends. Ligated fragments were then amplified for 12 cycles using primers incorporating unique index tags. Fragments were sequenced on an Illumina HiSeq 3000 using single end reads extending 50 bases. Basecalls and demultiplexing were performed with Illumina's bcl2fastq software and a custom python demultiplexing program with a maximum of one mismatch in the indexing read. RNA-seq reads were then aligned to the *Mus musculus* Ensembl release 76 top-level assembly with STAR version 2.0.4b (*Dobin et al., 2013*). Gene counts were derived from the number of uniquely aligned unambiguous reads by Subread:featureCount version 1.4.5 (*Liao et al., 2014*). Isoform expression of known Ensembl transcripts were estimated with Sailfish version 0.6.13 (*Patro et al., 2017*). Sequencing performance was assessed with RSeQC version 2.3 (*Wang et al., 2012*).

All gene counts were then imported into the R/Bioconductor package EdgeR (*Robinson et al., 2010*) and TMM normalization size factors were calculated to adjust for samples for differences in library size. Ribosomal genes and genes not expressed in at least two samples greater than one count-per-million were excluded from further analysis. The TMM size factors and the matrix of counts were then imported into the R/Bioconductor package Limma (*Ritchie et al., 2015*) and weighted likelihoods based on the observed mean-variance relationship of every gene and sample were then calculated with Limma's voomWithQualityWeights (*Liu et al., 2015*). The data were then fitted to a Limma generalized linear model to test for changes in a single cell population relative to the mean of all other populations to find only genes that were uniquely up-regulated with Benjamini-Hochberg false-discovery rate adjusted p-values less than or equal to 0.05. For each group of cells, global

perturbations in known Gene Ontology (GO) terms were then measured using the R/Bioconductor package GAGE (*Luo et al., 2009*) to quantify the mean log two fold-changes of ll genes in a given term versus the background log two fold-changes of all genes found outside the respective term. Only globally up-regulated GO terms with Benjamini-Hochberg false-discovery rate adjusted p-values less than or equal to 0.05 were considered for comparison across populations.

## Acknowledgements

We thank the Musculoskeletal Histology Core of Musculoskeletal Research Center supported by the National Institutes of Health P30 Grants AR057235 and P30 AR074992, and the Siteman Flow Cytometry Core at Washington University in St Louis. We also thank the Genome Technology Access Center in the Department of Genetics at Washington University School of Medicine for help with genomic analysis. The Center is partially supported by NCI Cancer Center Support Grant #P30 CA91842 to the Siteman Cancer Center and by ICTS/CTSA Grant# UL1TR002345 from the National Center for Research Resources (NCRR), a component of the National Institutes of Health (NIH), and NIH Roadmap for Medical Research. This publication is solely the responsibility of the authors and does not necessarily represent the official view of NCRR or NIH.

Data analysis was performed in part through the use of Washington University Center for Cellular Imaging (WUCCI) supported by Washington University School of Medicine, The Children's Discovery Institute of Washington University and St. Louis Children's Hospital (CDI-CORE-2015–505 and CDI-CORE-2019–813) and the Foundation for Barnes-Jewish Hospital (3770 and 4642).

We thank Danielle Ketterer, Sahil Mahajan PhD, Giulia Leanza PhD and Kristann Magee for the technical support in collecting in vivo samples.

This research was supported by grants from National Institutes of Health Grants R01 AR066551 (to RF), R01 CA235096 (to RF), grant from Shriners Hospital P19-07408 CR (to RF) and grants from Siteman Investment Program (to RF, RF and RC).

Partial support to this project was provided by a Just-in-Time Award by the Washington University Institute for Clinical and Translational Sciences (to RC and FF), Washington University (supported by NIH CTSA UL1 TR000448); and by funds by the Barnes-Jewish Hospital Foundation (to RC).

## Additional information

### Funding

| Funder | Grant reference number | Author |
| --- | --- | --- |
| National Institutes of Health | R01 CA235096 | Roberta Faccio |
| National Institutes of Health | R01 AR066551 | Roberta Faccio |
| Shriners Hospital | P19-07408CR | Roberta Faccio |
| Siteman Cancer Center | | Roberto Civitelli<br>Roberta Faccio |
| JIT Award by Washinghton University Institute for Clinical and Translational Sciences | TR000448 | Francesca Fontana<br>Roberto Civitelli |

The funders had no role in study design, data collection and interpretation, or the decision to submit the work for publication.

### Author contributions

Biancamaria Ricci, Conceptualization, Resources, Data curation, Formal analysis, Validation, Investigation, Writing - original draft; Eric Tycksen, Data curation, Software, Formal analysis; Hamza Celik, Resources; Jad I Belle, Methodology; Francesca Fontana, Resources, Funding acquisition, Validation; Roberto Civitelli, Conceptualization, Resources, Funding acquisition, Writing - review and editing; Roberta Faccio, Conceptualization, Resources, Data curation, Supervision, Funding acquisition, Writing - original draft, Project administration, Writing - review and editing

## Author ORCIDs

Biancamaria Ricci https://orcid.org/0000-0003-1574-6043
Roberto Civitelli http://orcid.org/0000-0003-4076-4315
Roberta Faccio https://orcid.org/0000-0003-1639-2005

## Ethics

Animal experimentation: This study was performed in accordance with the recommendations in the Guide for the Care and Use of Laboratory Animals of the National Institutes of Health. All the animals were handled according to the Institutional Animal Care and Use Committee (IACUC) protocols (#2016-0228 and # 2019-0982 to RF, #2014-0279 and #2017-0095 to RC).

## Decision letter and Author response

Decision letter https://doi.org/10.7554/eLife.54659.sa1
Author response https://doi.org/10.7554/eLife.54659.sa2

## Additional files

### Supplementary files

• Transparent reporting form

### Data availability

Sequencing data are deposited in GEO under accession code GSE143586 https://www.ncbi.nlm.nih.gov/geo/query/acc.cgi?acc=GSE143586.

The following dataset was generated:

| Author(s) | Year | Dataset title | Dataset URL | Database and Identifier |
|---|---|---|---|---|
| Faccio R, Ricci B, Tycksen E, Civitelli R, Fontana F, Celik H, Belle JI | 2020 | GSE143586 | https://www.ncbi.nlm.nih.gov/geo/query/acc.cgi?acc=GSE143586 | NCBI Gene Expression Omnibus, GSE143586 |

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
