## [Decision Letter]

**Acceptance summary:**

Your paper provides strong evidence that Osterix is expressed beyond the mesenchymal lineage and mark a set of early hematopoietic progenitors including CD 45 subsets of tumor promoter populations.

**Decision letter after peer review:**

Thank you for submitting your article "Osterix marks distinct subsets of CD45- and CD45+ populations in extra-skeletal tumors with pro-tumorigenic features" for consideration by *eLife*. Your article has been reviewed by three peer reviewers, and the evaluation has been overseen by a Reviewing Editor and Clifford Rosen as the Senior Editor. The reviewers have opted to remain anonymous.

The reviewers have discussed the reviews with one another and the Reviewing Editor has drafted this decision to help you prepare a revised submission.

Summary:

Ricci et al., present a study on dissecting a population of Cancer-associated fibroblasts (CAF). They utilized osterix-Cre/TdTomato lineage tracing approach to identify subpopulations of CAF that are fibroblastic in nature versus hematopoietic/immune cells. These population are distinct and they promote tumor expansion. In addition, they provide evidence that osterix is expressed very early in hematopoietic stem cells as proven by lineage tracing and absence of the mRNA transcripts for Osterix in mature hematopoietic cells that are labeled as Osx/TdTomato+ cells. Although comprehensive in nature, authors should provide additional evidence to strengthen the main hypothesis. This manuscript is well written, the data presented are extensive and novel. Its findings should be interesting for both the fields of stem cells, tumor and bone biology. However, this study does not challenge the dogma that osteogenic and hematopoietic markers are exclusive. But the study proves that Osterix is expressed in hematopoietic lineage and only at early stage of HSC stem/progenitor cells. There is no evidence provided that CD45+ cells are fibroblastic or osteogenic, and therefore the main focus is "non-specific" expression of osterix-Cre in CD45+ population. The paper challenges dogma that Osx is expressed only in mesenchymal cells, as indicated in the Discussion.

Essential revisions:

– Reviewers agree that a more definitive evidence should be provided that Osx transcripts/protein are expressed in CD45+ cells at a single cell level with an appropriate positive control, (including immunocytochemistry for OSX in TdT+ cells) and/or in situ hybridization. Maybe confocal imaging of CD45 immunostaining is required on the samples from tumors from OsxCre/TdTomato mice to assure that CD45 cells are not just attached to larger Osx-Cre/TD tomato;CD45- population.

– It is concerning that the number of Osx-cre/TdT+ cells increase drastically in a distant location of bone marrow after inoculating tumors. Do Osx-creER/TdT+ cells increase similarly in bone marrow in tumor bearing mice? An alternative explanation would be that systemic factors released from extra-skeletal tumors can enhance tet-transactivator or its operon activities in an Osx promoter independent manner, leading to increased cre expression. These technical possibilities should be carefully ruled out in the revised manuscript.

– To confirm that this osterix expression is not an artefact of particular transgenic line, authors have utilized sp7CreER^T2^ mice and presented in SF2, consider expanding data on this line, and moving it to a primary figure section.

– To better define the phenotype of the TdT-Osx+ cells additional markers of the osteoblastic lineage such as TNAP, BSP and Runx2 should be analyzed in Figure 2.

– To strengthen the point that the TdT-Osx+ cells are different in tumor-injected compared to naïve specimen, the latter should be included in Figure 2.

– Figure 4 should include no tumor controls for the analysis of bone marrow specimen.

---

## [Author Response]

Summary:Ricci et al., present a study on dissecting a population of Cancer-associated fibroblasts (CAF). They utilized osterix-Cre/TdTomato lineage tracing approach to identify subpopulations of CAF that are fibroblastic in nature versus hematopoietic/immune cells. These population are distinct and they promote tumor expansion. In addition, they provide evidence that osterix is expressed very early in hematopoietic stem cells as proven by lineage tracing and absence of the mRNA transcripts for Osterix in mature hematopoietic cells that are labeled as Osx/TdTomato+ cells. Although comprehensive in nature, authors should provide additional evidence to strengthen the main hypothesis. This manuscript is well written, the data presented are extensive and novel. Its findings should be interesting for both the fields of stem cells, tumor and bone biology.

We thank the reviewers for appreciating the novelty and the significance of the study. We performed key experiments as suggested by the reviewers to strengthen the main observation that OSX marks CD45+ populations (please see responses below to essential revision) and is expressed in subsets of HSCs.

However, this study does not challenge the dogma that osteogenic and hematopoietic markers are exclusive. But the study proves that Osterix is expressed in hematopoietic lineage and only at early stage of HSC stem/progenitor cells. There is no evidence provided that CD45+ cells are fibroblastic or osteogenic, and therefore the main focus is "non-specific" expression of osterix-Cre in CD45+ population. The paper challenges dogma that Osx is expressed only in mesenchymal cells, as indicated in the Discussion.

We absolutely agree with this comment and removed the sentence “this study challenges the dogma that osteogenic and hematopoietic markers are exclusive” and changed to “this study challenges the dogma that Osx is expressed only in mesenchymal cells”.

Essential revisions:– Reviewers agree that a more definitive evidence should be provided that Osx transcripts/protein are expressed in CD45+ cells at a single cell level with an appropriate positive control, (including immunocytochemistry for OSX in TdT+ cells) and/or in situ hybridization. Maybe confocal imaging of CD45 immunostaining is required on the samples from tumors from OsxCre/TdTomato mice to assure that CD45 cells are not just attached to larger Osx-Cre/TD tomato;CD45- population.

We performed immunofluorescence staining in TdT+ HSCs isolated by FACS sorting from naïve TetOFF;OsxCre;TdTomato (doxy treated) mice and demonstrate Osx nuclear staining at a single cell level (new Figure 8). Bone marrow stromal cells cultured in osteogenic media were used as positive control for osteoblasts and whole bone marrow from 8 weeks old mice, showing presence of few Osx+ and several Osx- cells, was used as an additional control.

We also provide additional data showing that TdT+CD45+ cells do exist and are not an artifact related to CD45+ cells being just attached to larger TdT+CD45- cells. We performed immunohistochemistry and immunofluorescence for CD45 on primary B16 tumor sections and bone marrow cells from DOXY-treated Osx-Cre;TdT animals. Images in new Figure 4 and Figure 4—figure supplement 2 show presence of single CD45+ cells (green membrane), single TdT+ cells (red cytoplasm) and in overlay presence of single CD45+TdT+ cells (green membrane + red cytoplasm).

– It is concerning that the number of Osx-cre/TdT+ cells increase drastically in a distant location of bone marrow after inoculating tumors. Do Osx-creER/TdT+ cells increase similarly in bone marrow in tumor bearing mice? An alternative explanation would be that systemic factors released from extra-skeletal tumors can enhance tet-transactivator or its operon activities in an Osx promoter independent manner, leading to increased cre expression. These technical possibilities should be carefully ruled out in the revised manuscript.

Reprogramming of hematopoiesis in the bone marrow is observed during tumor progression with a switch towards myelopoiesis. We and others have previously demonstrated increased immature myeloid populations in bone marrow of animals bearing B16-F10 tumors (Capietto et al., 2013; Meyer et al., 2018). Similarly, bone marrow residing MSCs also increase during tumor progression and higher numbers of osteoblast precursors have been reported in lung cancer (Engblom et al., 2017). Thus, the increase in Osx-cre;TdT+ cells in bone marrow of animals bearing a primary tumor was expected and is consistent with previous observations.

To further allay the concern that systemic factors released from extra-skeletal tumors can enhance Tet-transactivator in the TET-OFF;OsxCre/TdT mice, we analyzed the % of Osx-cre/TdT+ cells in bone marrow of OsxCreER^T2^;TdT animals. Tam was administered at 8 weeks of age, 3 days before tumor injection and then again 1, 6 and 9 days thereafter. Bone marrow was collected and analyzed on day 14. The relative % of TdT+ cells (including CD45- and CD45+ subsets) to total bone marrow cells from tumor bearing OsxCreER^T2^;TdT animals is significantly higher than in naïve mice (Figure 3E). These observations in Osx-creER^T2^/TdT mice, where Cre is driven by a promoter that does not contain the TET operon, along with direct evidence of Osx protein expression at a single cell level in HSCs, make it highly unlikely that TdT expression in Osx-Cre mice is independent of the Osx promoter. We have included this point in the Discussion.

– To confirm that this osterix expression is not an artefact of particular transgenic line, authors have utilized sp7CreER^T2^ mice and presented in SF2, consider expanding data on this line, and moving it to a primary figure section.

While we were eager to expand our data using the OsxCreER^T2^;TdT mouse line, our laboratory has been on a locked down for over 2 months and we had to drastically reduce our mouse colony, thus limiting the number of OsxCreER^T2^;TdT mice available to us for experiments.

Nevertheless, we were able to secure a cohort of mice to confirm an increase in TdT+ cells in bone marrow of tumor bearing mice compared to no tumor controls (new Figure 3, new Figure 4—figure supplement 1) and show presence of TdT+ HSC subsets (new Figure 7).

– To better define the phenotype of the TdT-Osx+ cells additional markers of the osteoblastic lineage such as TNAP, BSP and Runx2 should be analyzed in Figure 2.

We added analysis of TNAP, BSP and Runx2 as requested (new Figure 2).

– To strengthen the point that the TdT-Osx+ cells are different in tumor-injected compared to naïve specimen, the latter should be included in Figure 2.

We included RT-PCR analysis of TdT-Osx+ cells from naïve mice isolated from the bone marrow (new Figure 3—figure supplement 2).

– Figure 4 should include no tumor controls for the analysis of bone marrow specimen.

No tumor controls are shown in Figure 3C, D, E.